# The spatially heterogeneous and double-edged effect of the built environment on commuting distance: Home-based and work-based perspectives

**Zhong Zheng[1,2,3], Suhong Zhou[2]\*, Xingdong Deng[3]**

**1** Center for Territorial Spatial Planning and Real Estate Studies, Beijing Normal University, Zhuhai, Guangdong, P. R. China, **2** School of Geography and Planning, Guangdong Provincial Engineering Research Center for Public Security and Disaster, Sun Yat-sen University, Guangzhou, Guangdong, P. R. China, **3** Guangzhou Urban Planning & Design Survey Research Institute, Guangzhou, Guangdong, P.R. China

\* eeszsh@mail.sysu.edu.cn

**Data Availability Statement:** All relevant data are within the paper and its Supporting information files, except for the raw individual data.

## Abstract

Rich literature has examined the impact of the built environment on commuting distance. Linear models assume that the influence of the built environment is spatially homogeneous. However, given the spatial heterogeneity of urban space, conclusions might be different or even be contrary. The influence of the built environment might also be different by home and work locations. To explore the spatially heterogeneous effect of the built environment from both home-based and work-based perspectives, this study applied large-scale cellular cellphone data in Guangzhou, China. Commuting was measured by decay parameters of probabilistic distributions of commuting distances. Geographically weighted regression models were applied to examine the spatially heterogeneous effect, differentiated by home-based and work-based perspectives. Results confirmed that the impact of the built environment on commuting distance is spatially heterogeneous. The urban space is classified into clusters of central areas, inner suburbs, and outer suburbs. Results also revealed the double-edged effect of the built environment. Residential population, recreation facilities, and mixed development are residence-attractive factors that increase the home-based commuting distance and decrease the work-based commuting distance. Work population and transport facilities are work-attractive factors that decrease home-based commuting distance and increase work-based commuting distance. The results further provide evidence to support area-based policies in urban planning practice.

## Introduction

Long commuting distance causes problems of traffic congestion, air pollution, and car dependence [1]. Rich literature has examined the impact of the built environment on commuting to provide planning suggestions for policymakers. Several built environment factors—such as the

**Funding:** ZZ received some funding for this work from National Natural Science Foundation of China (http://www.nsfc.gov.cn/english/site_1/index.html, grant number 42101223), The Ministry of education of Humanities and Social Science project (http://www.moe.gov.cn/s78/A13/, grant number 20YJC630232), and Guangdong University Innovation Team Project (https://210.76.75.91/indexAction!to_index.action, grant number 2021WCXTD014). SZ received funding for this work from National the Natural Science Foundation of China (http://www.nsfc.gov.cn/english/site_1/index.html, grant number71961137003).

**Competing interests:** The authors have declared that no competing interests exist.

relation of jobs and housing, mixed land use, commercial development and infrastructure provision—are found to be related to commuting. However, there are still debates on how they affect commuting in a city-wide spatial context. For example, under the co-location hypothesis, the co-location of housing and jobs is associated with shorter commuting distance [2]; commercial land use decreases commuting distance in a mixed land use neighborhood, and mixed land use encourages non-vehicle trips [3]. However, some authors have different opinions. In tradition urban structure models (i.e. Burgess Model [4], Alonso Model [5]), it is natural to see imbalanced jobs and housing in industrial agglomeration areas. The co-location of housing and jobs would not significantly reduce commuting distance [6]. And mixed land uses have no significant impact on commuting distance [7].

The debate on the impact of the built environment, in our opinion, is mainly caused by the different mechanisms of human and urban space interaction: the market mechanism and the individual choice mechanism. The economic agglomeration effect triggered by the market mechanism shapes the urban spatial structure deeply. Economic agglomeration refers to a large number of firms existing in spatial proximity and benefit from cost reductions and efficiency gains [8]. It encourages capital facilities and buildings to be concentrated located [9]. The individual choice mechanism means that a decision maker chooses the residential and work location with the highest utility [10]. It assumes that workers choose home locations as close to their jobs as possible [6]. The two mechanisms have different impact on different urban locations. The market mechanism is more competitive at business centers, and the choice mechanism has stronger influence at residential and suburban areas [4]. Since the urban space is heterogeneous, the relationship between the commuting distance and the built environment should be spatially varied [11,12]. Most studies of commuting and built environments are based on linear models (or global models), which assume that the impact of the built environment is spatially homogeneous. However, their relations in a city are naturally heterogeneous due to the spatially varied effect of the market mechanism and the individual choice mechanism. For example, commuting patterns in a central area and a suburban area are different. Based on different theoretical framework, it is not surprising to see that the relationships between commuting and the built environment are different from study to study. From a geographical perspective, the Tobler's first law of geography [13] assumes that near things are more related than distant things. It causes locational effects [14] that, for example, residents sharing the same range of geographical environments are likely to have similar and localized commuting behavior. More importantly, variables describing the heterogeneity of spatial attributes are often absent or cannot be obtained by researchers [15]. Hence, researchers should consider the spatially heterogeneous effect. It helps extend the understanding of commuting and built environment relations from a global context to a spatially varied and localized context.

Geographically weighted regression (GWR) is a spatial statistical model. It reveals geographical variations in the relationship between a response variable and a set of covariates [16]. The model estimates a set of spatially varying coefficients, which can capture heterogeneous effects. It is different from a 'global' linear regression model which estimates an averaged single coefficient value across the entire study area. Rather, GWR is a 'local' model that exhibits complex correlations in different areas. Again, the localized correlations are based on similar behavior of individuals who share the same range of spatial contexts.

In aggregate analysis, the commuting distance of a spatial unit is generally measured by the average value of all travelers' commuting distances within that spatial unit [17–21]. It is important to note that, for the same spatial unit or neighborhood, there are two ways of averaging the commuting distance: as a home-based measure and as a work-based measure. The home-based measure calculates the average travel distance of commuters who depart from the spatial

unit, while the work-based measure is based on commuters who arrive at the spatial unit. Because the results of the two measurements are different, it is necessary to differentiate between home-based and work-based commuting distances. More importantly, the underlying mechanisms are different. From a work-based perspective, the economic agglomeration is the dominant mechanism. Industrial firms have much stronger land-rent bidding ability than individuals in a free market system. Local workers are forced to live far from workplaces [22]. From a home-based perspective, the co-location theory is the dominant mechanism. Workers can freely choose their home locations to save commuting time where the supply of housing land is adequate. Current studies are limited in not considering the home-based and work-based perspectives simultaneously. Analysis of the aggregate commuting distance based on the home location is mainstream in the literature [17–21] since a travel survey is normally conducted at home locations. However, few studies have analyzed the built environment's impact on both home-based and work-based commuting distances. In this study, an underlying hypothesis is that the impact of the built environment on home-based and work-based commuting distances may be different or even contrary. It causes a double-edged effect. To examine the double-edged effect, it is necessary to analyze the relationship between commuting distance and the built environment from both the home-based and work-based perspectives.

To address the research gaps, the paper explores the spatial heterogeneous and double-edged effect of the built environment on the home-based and work-based commuting. It develops a new method to explore the commuting pattern of a whole city using cellphone data. Commuting is represented by a decay parameter of the probability distribution of commuting distances. Geographically weighted regression models are applied to investigate the spatially heterogeneous impact of the built environment. The double-edged effect is examined by the different impacts of the built environment on home-based and work-based trips. The conceptual framework is presented as Fig 1. It assumes that there are two mechanisms which dominate the relationship between the built environment and the commuting distance: the market mechanism and the individual choice mechanism. The relationships are varied at different urban locations, causing spatially heterogeneous effect. Also, the market mechanism and the individual choice mechanism are the leading force of the work-based and home-based commuting, respectively. According to the conceptual framework, there are four research questions:

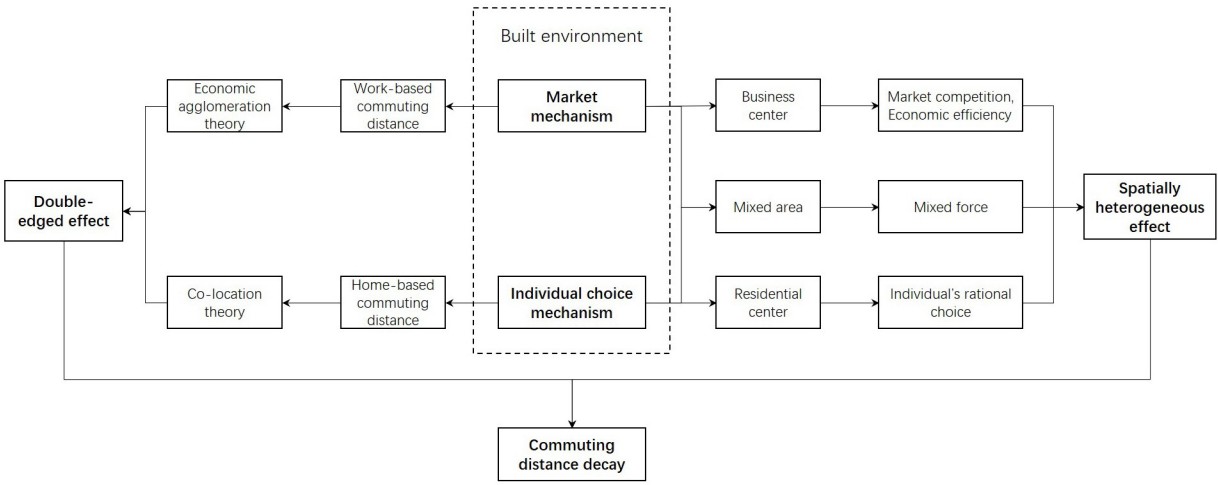

**Fig 1. Conceptual framework.**

**Questions 1 & 2**. What is the built environment's spatially heterogeneous impact on home-based and work-based commuting distances?

**Question 3**. Does the double-edged effect exist based on the built environment's different impacts on home-based and work-based commuting distance? To what extent does it influence commuting distance?

**Question 4**. Based on the spatial heterogeneous and double-edged effect, how can the spatial pattern of the built environment's impact be summarized? And what planning strategies can we develop from the spatial pattern?

By answering these research questions, the paper contributes to current research in two ways. First, it examines the spatially heterogeneous effect of the built environment, which provides a new understanding of the built environment and commuting relationship. Second, it reveals the double-edged effect of the built environment's impact. Research findings can be further applied to develop zonal planning policies for government.

## Literature review

### The relation between commuting distance and the built environment

The commuting distance and built environment relationship is traditionally analyzed from either the home-based or work-based perspective. Most studies are conducted at home locations. Several built environment factors, such as jobs and housing, recreation facilities, public transport, and mixed land use, have been found to be associated with commuting distances. The impact of the jobs–housing relationship is the issue of most concern. Most studies confirmed that the co-location of jobs and housing would shorten the commuting distance. Peng [23] investigated the relationship between VMT [vehicle miles traveled] and the jobs–housing ratio in Portland. VMT significantly changes when the jobs–housing ratio is less than 1.2 or larger than 2.8. Levinson [24] used data from 8000 household travel surveys in Washington DC to analyze the accessibility to work. It was found that residences in job-rich areas and workplaces in housing-rich areas are associated with shorter commutes. Sultana [25] examined the relationship between the average commuting time and job–housing ratio and house price using census data. The results confirmed that the imbalance between the locations of jobs and housing is the dominant factor in long commuting. Zhao et al. [21] used travel survey data in Beijing to investigate the relationship between commuting time and jobs–housing balance. Jobs–housing distance reduces the commuting time significantly. Results have also shown that workers living in Danwei housing (housing provided by employers) have shorter commuting time. Lin et al. [18] analyzed the relationship between commuting and job–housing ratio, social-demographic variables and transport modes in Beijing. The results showed that jobs–housing balance has significant influence on commuting time, and that commuting behavior is strongly related to income, gender, age, education and land use reform. Applying aggregate analysis to travel survey data in Gauteng, South Africa, Geyer and Molayi [26] examined the relationship between average travel time and the job–employed ratio, internal capture ratio and other social-demographic variables. They found that workers have higher average travel time in job-rich and balanced areas. These studies confirmed that the co-location of jobs and housing would reduce commuting distance or time. However, some studies have contrary findings that a jobs–housing balance does not significantly influence commuting. Self-selection related to housing location preference is the key factor rather than the jobs–housing balance, and a jobs–housing balance would not significantly decrease the commuting distance at

the sub-area level [27]. Giuliano [6] argued that jobs–housing balance cannot effectively solve the traffic congestion problem since the relationship between jobs and housing is complex especially in multi-worker households.

Commercial and recreation facilities have also been found to be related to commuting. Gordon et al. [28] found a significant correlation between commuting times and commercial densities. Cervero [3] found that commercial land use near housing is associated with short commuting distances and low vehicle ownership. However, the authors in another study argued that jobs–housing balance is a more direct method to reduce travel than retail–housing mix [29].

Public transport service is normally believed to be an effective way to encourage non-vehicle travel [30]. However, new findings from Atlanta suggest that a new subway expansion would increase commuting trips, and non-vehicle trips would not be reduced as expected [31]. Song et al. [32] found that the choice of public transport is positively associated with commuting times, which suggests the need to provide a high-quality public transport system.

Land use mixture measures the diversity of urban space. It is assumed that mixed land use is associated with fewer trips [33]. Several studies have confirmed the assumption. For example, trip lengths are shorter at locations with mixed uses [34], and commuters living in mixed land use neighborhoods travel shorter distance [35]. However, it is also argued that a retail–housing mix does not reduce trips as much as the jobs–housing balance does [29].

Compared to home-based analysis, there is limited work-based analysis of the commuting distance and built environment relationship. Taking an industrial park as a case, Zhou et al. [22] found that excess commuting is correlated with the oversupply of industrial land and shortage of public and residential land, high housing price and increasing vehicle travel. Conducting travel surveys in universities, researchers found that employees with a large employer and higher income have a better jobs–housing balance. Lower income employees have to commute long distance to find lower housing prices.

## Spatial effect of built environment on commuting

Recently, some researchers have noted that the impact of the built environment on commuting may not be linear. Rather, the spatial effect plays a large role in the relationship between the built environment and commuting because of spatial dependency, spatial heterogeneity and spatial heteroscedasticity [15]. Spatial heterogeneity refers to variations in relationships between the dependent (commuting distance, travel mode) and independent variables (built environment, social demographics) across spatial units. Ignoring the spatial effect would cause inconsistent parameter estimation because a single linear model can only 'averagely' reflect the global relation but not any local part of the relation [15]. Some researchers have already realized the problem. Taking a city or a part of a city as cases, studies of the heterogeneous impact of the built environment can be summarized in three aspects.

First, the urban space is roughly differentiated by central and suburban areas. The underlying hypothesis is that central and suburban areas have significantly different spatial contexts that affect commuting behavior. To a large extent, the difference relates to urban spatial structure, particularly the decentralization trend in city development. The trend of low density, dispersed suburbanization and decentralization in urban spatial structure could lead to either an increase or decrease in the average commuting distance [36]. Some studies have observed an increasing trend in commuting distance from a dispersed urban form. For example, the suburbanization of jobs is associated with increasing congestion, increasing trip lengths, and more work trips [24]. The shift from a monocentric to a dispersed city form increases commuting time [37]. A polycentric city model increased urban commuting more than a monocentric

model [38]. However, other authors have contrary findings. The alternative of job relocation can significantly affect commuting travel savings. The spatial distribution of jobs should be decentralized to respond to the dispersed population distribution. The discussion on decentralization and suburbanization implies the spatial heterogeneous impact of the built environment. A noticeable difference is between city centers and suburbs [39]. Self-containment of employment is significantly affected by the jobs–housing balance in the suburbs but has limited effect in central areas [40]. Residents with better proximity to an employment sub-center and better subway accessibility would tend to travel shorter distance [35]. Shorter commuting is related to accessibility and increased residential land use at employment centers, and more jobs in public transport corridors [19]. Research on transport emissions has found that the impact from the built environment, such as residential density, entropy and intersection density, is significant in both urban and suburban areas, with residential density having more impact on suburban than urban areas [41]. These studies simply divide the urban space into two types of central areas and suburban areas, but a city is a far more complex system. A sophisticated classification is needed to reflect the nature of the urban space.

A better solution is to use a multi-level model, which is the second aspect of related studies. A multi-level model has a hierarchical structure with an individual level and a spatial level. The heterogeneous spatial effect is captured by the spatial level. It assumes that parameters vary by groups of people who locate in the same spatial unit. The urban space is no longer simply differentiated by centers and suburbs. Rather, the multi-level method can reflect the heterogeneity across different places. Applying a multi-level mixture hazard model, the spatial effect of the built environment's influence on commuting distance was stressed, since parameters are significantly heterogeneous. The authors suggested that spatial heterogeneity should be further analyzed by a spatial model when considering the spatial autocorrelations effect or the horizontal spatial dependence among different locations [14]. Wu and Hong [31] similarly believe that using spatial models is important to analyze the relationship between urban form and travel behavior, because the influence of the built environment varies among different locations. Applying the multi-level analysis framework, the spatial heterogeneous effect of the built environment on car ownership was investigated in Maryland and Washington DC. It has been found that the built environment explains 42.8% of the spatial heterogeneity in household car ownership [42].

However, the multi-level model has a strong assumption that the impact of the built environment is homogeneous within the same spatial unit and heterogeneous among different spatial units. The multi-level model is especially suitable for studies based on travel surveys in which respondents are selected from several nonadjacent sampling places. Respondents living in the same neighborhood are assumed to be similarly affected by the built environment. However, it is also possible that a travel survey is conducted in all spatial units or census blocks. When dealing with spatial heterogeneity across the entire study area, the multi-level model has to imply the existence of a boundary and divide the study region into several sub-regions. It is doubtful whether and how boundaries exist, and there is also the 'modifiable areal unit problem' [43]. More importantly, the nature of spatial heterogeneity does not mean that spatially related individuals always have similar behavior. In contrast, it is also possible that near things are not alike because of negative spatial auto-correlation [44]. Therefore, new models are needed to describe spatially varying effects with non-predefined sub-regions.

The solution is the third domain of literature, a geographically weighted regression (GWR) model. The model applies a 'local' form of spatial statistical analysis to estimate a set of spatially varied parameters. It reflects the geographical variations in the relationship between a dependent variable and an independent variable [16]. It contrasts with a 'global' model or a linear regression model which estimates a unique parameter across the entire study area. The spatial

variations in relationships between the built environment and commuting successfully capture the spatial heterogeneous effect [44]. The GWR model was applied to examine the spatially heterogeneous impact on land prices in Beijing, China. Results confirmed that a spatial model better reflects the nature of the land market than a non-spatial model, and there is a heterogeneous linkage between government-funded amenities and land prices [45]. Similarly, a study explored the heterogeneous relationship between transport accessibility and land value in the Tyne and Wear region in the UK. It was found that transport accessibility has a double-edged effect on the land value, with a positive impact in some areas but a negative impact in others [40]. The finding enhances the importance of the spatially heterogeneous effect compared to a global model [46]. Zhang et al. [47] used a multi-scale geographically weighted regression model to examine the spatial interaction of expressway transport flows in Jiangsu Province, China. It illustrates the spatial effects at varied scales between push and pull forces of express trips at a regional scale. Applying a GWR model, a study examined the spatially heterogeneous effect of the built environment on parking in Shenzhen. Results demonstrated that floor area ratio has a larger increasing effect in suburban areas, lot size has a stronger positive impact in areas with higher parking demand, and the impact from transit accessibility is inconsistent across the whole city [48].

In summary, the spatially heterogeneous or spatially varying effect is based on the locally similar behavior of individuals who share the same range of spatial contexts. Related studies have investigated the spatial effect indirectly or directly. However, the spatial heterogeneous effect in aggregated commuting behavior is still unclear. The heterogeneous effect should be revealed explicitly at a finer spatial scale since commuting is generally a city-wide issue. Therefore, it is necessary to apply a local model, a geographically weighted regression model, to give an overview of the structural relationship between commuting and the built environment of the whole city. The analysis can be further used by urban planners or policymakers to optimize the spatial layout of urban functional zones [49].

## Study area and data

The model is tested using cellphone data from the inner city of Guangzhou, China (Fig 2). Guangzhou is one of the four first-tier cities in China and a provincial capital. The study area includes districts of Yuexiu, Tianhe, Haizhu, Liwan, Baiyun, Huangpu, and Panyu, the urbanized area but not the whole city. This area is of 2435.7 $km^2$ and a population of 11.5 million in 2019 (Guangzhou Statistics Bureau: http://tjj.gz.gov.cn/tjdt/content/post_5727607.html). The city's population is concentrated in the urbanized area such that 75% of the population lives in 32% of the area. We select it as the study case because the un-urbanized area is mainly rural and forest land with sparsely distributed residential settlements.

A Chinese mobile operator provided the cellphone data. It accounts for about 20% of the user market. One month of data (September 8, 2017 to October 8, 2017) in the study area are used. Signal towers record cellphone users' locations. A user has two possible status types: stay and movement. When a user stops at the same location for more than 1 hour, it is defined as stay. Otherwise, a user is in movement. When a user's stay location at night (11 pm–5 am) is the same location over 20 days, the location is defined as the user's residential place. Similarly, when a user's stay location in daytime (9 am–5 pm) is the same location over 20 days, the location is defined as the user's workplace. Movement between a residential location and a work location in the morning peak (7 am–9 am) is defined as a commute trip. To protect users' privacy, the number of users is counted by spatial cells defined as 500 m by 500 m. The cellphone dataset contains 13.7 million commuting trips in one month.

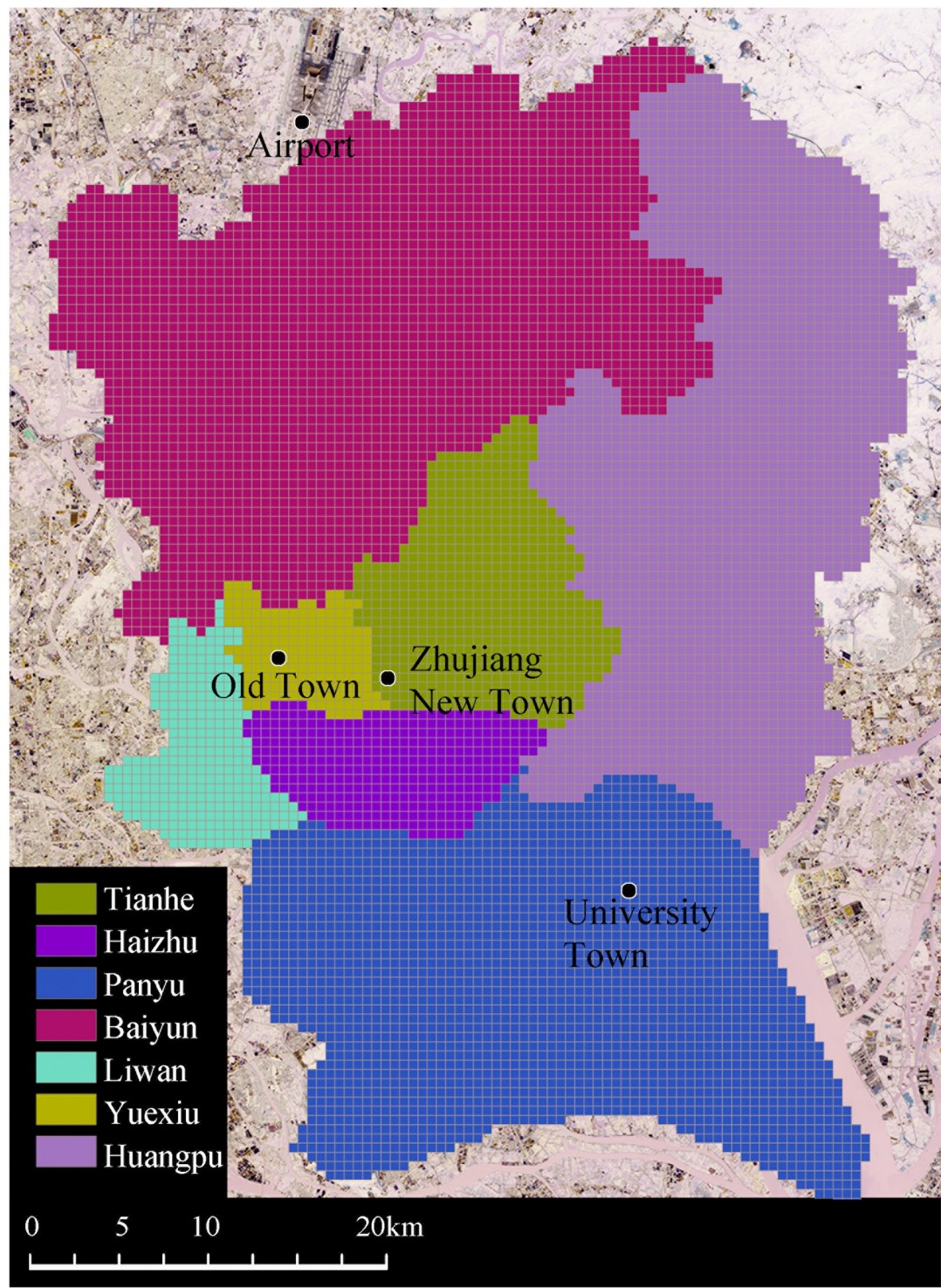

**Fig 2. Study area.**

## Method

### Measurement of the commuting distance

The commuting distance of a spatial unit is generally represented by the average travel distance of all commuters within that unit [23,26,50]. However, averaging the travel distances of all commuters into a single value leads to the loss of rich travel information. Instead, this study measures the commuting distance by the decay parameter of its probabilistic distribution. The distance decay parameter describes how the travel probability decreases with the increase of commuting distance in a spatial unit. Using the decay parameter to measure the commuting distance is advantageous in including all commuters' travel distance information. In this study, the commuting distance of all cellphone users follows an exponential distribution (see [51]). The probabilistic distribution of commuting distance, represented by the cumulative distribution function (CDF) of an exponential distribution, is:

$$P(X \leq x) = 1 - \exp\left(-\frac{x}{\beta}\right) \ (x \geq 0) \tag{1}$$

where $\beta$ is the decay parameter, and $P(X \leq x)$ is the probability (CDF) that the travel distance $X$ is less than value $x$, since the probability of a continuous random variable can only be expressed by a cumulative distribution function. $\beta$ is estimated by a maximum likelihood method. The likelihood is given by [52]:

$$L(\beta, x) = L(\beta, x_1, x_2, \ldots x_n) = \prod_{i=1}^{n} \frac{1}{\beta} \exp\left(-\frac{x_i}{\beta}\right) \tag{2}$$

$x_i$ is the observed commuting distance with n samples. $\beta$ is estimated by maximizing $L(\beta, x)$, and we get: $\beta = \bar{x}$. It implies that the decay parameter is the expectation of the distance distribution in an exponential distribution. Note that $\beta$ is not simply a mean of commuting distances in a spatial unit. Rather, we should treat it as a weighted average of that, according to the property of an exponential distribution. In other words, the expectation value also shapes the slope of the curve. It represents either a high probability of longer distance and a low probability of shorter distance (large $\beta$), or a high probability of shorter distance and a low probability of longer distance (small $\beta$).

After revealing the exponential distribution of commuting distance, the method is applied to each spatial cell separately (1601 cells in total). Each cell has a unique commuting distance distribution. The number of trips in each cell derived from the cellphone data is large enough to define a distance distribution. Aggregating departure trips at a location represents a home-based perspective, while aggregating arrival trips at a location represents a work-based perspective. The decay parameter is spatially heterogeneous since travel distance distributions vary across different locations.

### Variables

A geographically weighted regression (GWR) model is applied to examine the spatially heterogeneous impact of the built environment on commuting distance. The dependent variable is decay parameter $\beta$ in Eq 1. The decay parameters $\beta$ of commuting distance distributions are differentiated by departure trips and arrival trips, which represent home-based and work-based commuting distances respectively. Independent variables (Table 1) are selected according to the 'Ds' measurement [34] of the built environment: density, diversity, design, destination accessibility, and distance to transit. Points of interest (POIs) data are also applied to represent the built environment. A 'point of interest' data point records information about a

**Table 1. Description of independent variables.**

| Type | Independent variables | Mean | Minimum | Maximum | Std. Deviation |
|------|----------------------|------|---------|---------|----------------|
| Local | Residential population | 867.32 | 0 | 7360 | 832.44 |
| | Work population | 482.50 | 0 | 6019 | 555.02 |
| | Recreation POIs | 63.40 | 0 | 1046 | 90.30 |
| | Transport POIs | 7.01 | 0 | 269 | 12.82 |
| | POI mixture | 0.54 | 0 | 0.8306 | 0.25 |
| | Road intersections | 4.84 | 0 | 75 | 6.20 |
| | Bus stops | 2.08 | 0 | 22 | 2.66 |
| Global | Closeness | 0.000876 | 0 | 0.001304 | 0.000237 |
| | Distance to center | 13233.89 | 0 | 36458.9 | 7358.59 |

coordinate location and a functional type of a spatial facility from a navigation map. POI data are sourced from the Baidu map and provided by the Daodaotong company. A total of 27,349 POIs are used in this study.

This study does not include socioeconomic attributes, such as GDP, gender ratio, or elderly population ratio. The aim of this analysis is to explore the relation between the built environment and the commuting distance. By revealing the relation, the government can directly implement planning and policy measures for the built environment to address the problem of long commuting distance.

Density is generally measured by population which reflects the intensity of human activity. In this study, density refers to residential and work population. The impact of the jobs–housing relationship is the most concerning issue. Most studies have confirmed that the co-location of jobs and housing would shorten the commuting distance. For example, the analysis from Rivera and Tiglao [24] found that residences in job-rich areas and workplaces in housing-rich areas are associated with shorter commutes. Sultana [25] confirmed that the imbalance between jobs and housing locations is the dominant factor in long commuting. Following a similar approach, this study used the residential population and the work population as measures of density, which are identified from cellphone data.

Diversity is generally measured by the land use mixture. It assumes that mixed land use is associated with fewer commuting trips [33]. Several studies have confirmed the assumption. For example, trip lengths are shorter at locations with mixed uses [34], and commuters living in mixed land use neighborhoods would travel shorter distance [35]. However, it is also argued that mixing retail and housing does not reduce trips as much as the jobs–housing balance [29]. This study applied the mix of POIs instead. The advantage of the mix of POIs is that it considers the mixture of spatial facilities rather than land use. It is calculated by information entropy ($p_n$ is the percentage of the POI number with type $n$ of the total POI number in a cell):

$$H = -\sum_{n} p_n \log p_n \tag{3}$$

Diversity can also be measured by functional facilities. Diversity represents the degree to the land use difference represented by land area, floor area or employment [34]. Functional facilities such as recreation and transport facilities are found to be related to commuting. Results from Gordon et al. [28] revealed a significant correlation between commuting times and commercial facilities. It was also found that commercial land use near housing is associated with short commuting distance and low vehicle ownership [3]. Also, the provision of transport facilities such as parking and car services has been widely believed to be associated

with increasing vehicle commuting [48]. Hence, in this study the functional facilities are measured by the number of recreation POIs and transport POIs. Recreation POIs include dining (20.6%), public services (15.7%), entertainment (11.5%) and shopping (52.2%). Transport POIs include parking lots (51.6%), car services (34.0%) and important transport navigation spots such as toll gates, bridges and train or bus stations (14.4%). Transport facilities are mainly associated with vehicle trips.

Design is measured by the number of road intersections in a cell.

Distance to transit can be alternatively measured by the number of stations per unit area [34]. In this study, it is the number of bus stops in a grid.

Destination accessibility is measured by space syntax closeness and distance to the center. The concept of closeness is from the space syntax theory. It measures the centrality level of the road network. Closeness, also normalized as syntactic 'Integration' [53], is a key index of the centrality. It indicates the accessibility and centrality level of spatial units [54]. In other words, it measures the closeness of any given road section to all other road sections in the system [55]. As an index of the destination accessibility, it is advantageous in not necessarily predefining a center. A road with the highest closeness means that it is close to all roads in the study area, and it is the geometric center of the road network. Therefore, a location with higher closeness value has better destination accessibility. The space syntax closeness of the road network is calculated by:

$$c_i = \frac{N - 1}{\sum_{j=1}^{N} d_{ij}} \tag{4}$$

where $d_{ij}$ is the shortest distance between road section $i$ and $j$, $N$ is the total number of road sections in the study area. The closeness of road sections is aggregated into cells by:

$$C = \sum_i c_i l_i / \sum_i l_i \tag{5}$$

where $c_i$ is the closeness of road section $i$ in a cell, $l_i$ is the length of section $i$.

The distance to the center variable is the Euclidean distance of a cell to Zhujiang New Town CBD.

Built environment variables should be further differentiated by local variables and global variables in a GWR model. A local variable means its coefficient value varies across different locations, while a global variable has a unique coefficient value across the entire study area like a linear regression model. Variables not able to capture the spatial relation, such as residential population, work population, recreation POIs, transport POIs, POI mixture, road intersections and bus stops are local variables. The value of local variables varies across different locations. In contrast, variables of closeness and distance to the center are global variables since they represent the location effect. These variables themselves can describe the spatial relation to the city center. In addition, all variables are measured in a single cell without including adjacent cells. Since the GWR model itself is a spatially weighted algorithm considering the built environment's impact from adjacent cells, there is no need to measure the built environment from adjacent cells again. Spatial distributions of built environment variables are shown in Fig 3.

## Geographically weighted regression

A GWR model belongs to the regression model family, but its parameters are geographically varying. A typical GWR model is formulized by [16]:

$$y_i = \sum_k b_k(u_i, v_i) x_{ki} + \varepsilon_i \tag{6}$$

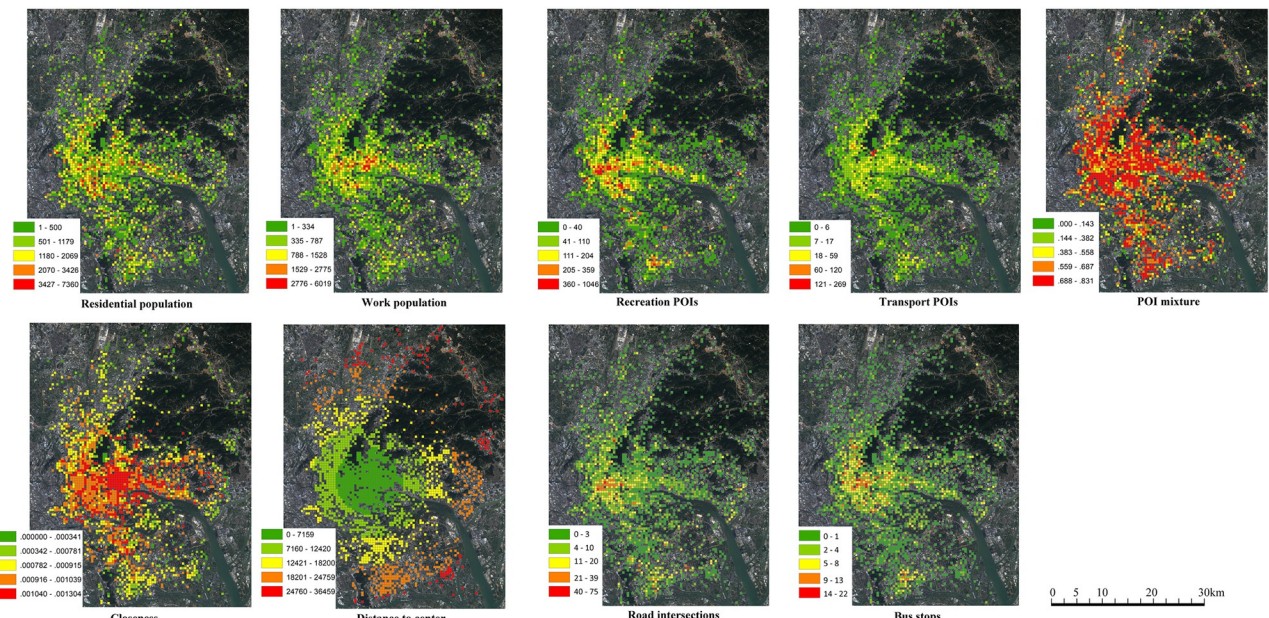

**Fig 3. Spatial distributions of built environment variables.**

where $i$ denotes a location, $y$ is the dependent variable, $x$ is the $k$th independent variable, and $\varepsilon_i$ is the Gaussian error, $(u_i, v_i)$ is the longitude and latitude coordinate; and coefficient $b_k(u_i, v_i)$ is a geographically varying-parameter defined by a weighting function. The concept of a GWR model is that the dependent variable at location $i$ interacts with independent variables of observations falling within a bandwidth of location $i$. An observation nearer to $i$ impacts estimating parameters of $i$ more than one farther away. The weighting function $b_k(u_i, v_i)$ reflects the distance decay effect. It is usually expressed by a Gaussian function:

$$\omega_{ij} = \exp\left[-\frac{1}{2}\left(\frac{d_{ij}}{B}\right)^2\right] \tag{7}$$

or a Bi-square function:

$$\omega_{ij} = \begin{cases} \left[1 - \left(d_{ij}/B\right)^2\right]^2 & j \in \{N_i\} \\ 0 & j \notin \{N_i\} \end{cases} \tag{8}$$

where $d_{ij}$ is the distance between location $i$ and $j$, $B$ is bandwidth. Bandwidth is the search range of the model. A cell is affected by all other cells within the range of the bandwidth. Bandwidth can be manually chosen or determined by criteria such as cross-validation or Akaike information criterion (AIC) [56]. The GWR model is applied to home-based and work-based commuting separately.

## Two-step cluster

To reveal the double-edged effect explicitly, a two-step cluster model is applied to further examine the area-based impact of the built environment and answer research question 3. The algorithm of the two-step cluster is defined in Rundle-Thiele et al. [57]. In brief, the first step

pre-clusters the data into several small sub-clusters using a cluster feature tree, and the second step aggregates sub-clusters into clusters using the standard hierarchical clustering algorithm. The clustering algorithm can generate different numbers of clusters. The optimal number of clusters is determined by Schwarz's Bayesian information criterion (BIC). In this study, built environment variables are transferred from continuous values into categorical values: positive, negative, and not significant. The categories of all variables from both home-based and work-based GWR models are the input to the two-step model. The model will identify clusters of spatial cells according to the similarity of the built environment's impact. It will reveal a new urban spatial structure in terms of urban space and transport relationships.

## Results of GWR models

This part answers research questions 1 & 2. The GWR model is firstly compared with an ordinary linear regression model. An ordinary linear regression model can be seen as a global result in a GWR model with spatially homogeneous coefficients. AIC is an indicator to test the performance of an ordinary linear regression model and a GWR model [58]. It tests both the accuracy and complexity of a model. This study used AICc instead. When the sample size is small, there is a probability that a model with too many parameters may have better AIC performance and the model is overfit. To address such potential overfitting, AICc was developed. AICc is AIC with a correction with small sample size such that $AICc = \text{AIC} + 2K(K + 1)/(n - K - 1)$ where $K$ is the number of parameters and $n$ is the number of observations [59]. When the difference of AICc value between the two models is greater than 3, the model with lower AICc is better. In this study, the AICc of the home-based model and the work-based model are both far less than ordinary linear regression models (Table 2). The $R^2$ also proves that GWR models fit better than ordinary linear regression models.

The coefficients of the home-based and work-based models are displayed in Table 3. The significance is tested at the 0.05 level in the t-test. A GWR model does not generate a specific value for each variable. Rather, it produces a series of coefficient values for local variables to represent the geographically varying effect. Table 3 clearly shows the quartile, median and value range of each variable for the home-based and work-based models. Interestingly, most variables have both positive and negative effects on commuting distance. This challenges previous findings on the built environment and commuting distance relationship. The result suggests that spatial heterogeneity should be considered. Global variables have a unique coefficient value because they are assumed to be not geographically varying. Closeness measures how much a location's road network is close to the center. Larger closeness is associated with shorter commuting distance in both home-based and work-based models. The distance to the center increases the commuting distance, indicating that people commute longer at a

**Table 2. GWR model results.**

| Home-based model | Linear regression | GWR |
|---|---|---|
| AICc: | -4776.435 | -5008.079 |
| R square | 0.0570 | 0.274 |
| Adjusted R square | 0.0519 | 0.195 |
| Work-based model | Linear regression | GWR |
| AICc: | -3381.227 | -3911.423 |
| R square | 0.233 | 0.540 |
| Adjusted R square | 0.229 | 0.457 |

**Table 3. Parameter estimation in GWR models.**

**Home-based model**

| Global coefficients | Estimate | Std. error | | | | | |
|---|---|---|---|---|---|---|---|
| Closeness | -0.014 | 0.0120 | | | | | |
| Dist. to center | 0.327 | 0.0177 | | | | | |
| Local coefficients | Mean | Std. Dev | Min. | Max. | Lower quartile | Median | Upper quartile |
| Intercept | 0.118 | 0.035 | 0.036 | 0.197 | 0.099 | 0.121 | 0.144 |
| Residential population | 0.057 | 0.113 | -0.474 | 0.369 | -0.010 | 0.032 | 0.122 |
| Work population | -0.272 | 0.320 | -1.446 | 0.087 | -0.418 | -0.175 | -0.035 |
| Recreation POIs | 0.053 | 0.127 | -0.535 | 0.495 | -0.022 | 0.023 | 0.110 |
| Transport POIs | 0.081 | 0.375 | -1.405 | 1.188 | -0.043 | 0.075 | 0.287 |
| POI mixture | 0.031 | 0.029 | -0.040 | 0.123 | 0.011 | 0.030 | 0.050 |
| Road intersections | -0.116 | 0.141 | -0.499 | 0.639 | 1.139 | -0.202 | -0.087 |
| Bus stops | 0.000 | 0.104 | -0.405 | 0.372 | -0.048 | 0.004 | 0.052 |

**Work-based model**

| Global coefficients | Estimate | Std. error | | | | | |
|---|---|---|---|---|---|---|---|
| Closeness | -0.0118 | 0.0160 | | | | | |
| Dist. to center | 0.0719 | 0.0303 | | | | | |
| Local coefficients | Mean | Std. Dev | Min. | Max. | Median | Lower quartile | Upper quartile |
| Intercept | 0.178 | 0.059 | 0.050 | 0.338 | 0.288 | 0.133 | 0.176 |
| Residential population | -0.205 | 0.289 | -1.912 | 0.592 | 2.505 | -0.295 | -0.154 |
| Work population | 0.360 | 0.396 | -1.000 | 2.492 | 3.492 | 0.214 | 0.344 |
| Recreation POIs | -0.154 | 0.221 | -1.628 | 0.652 | 2.280 | -0.259 | -0.122 |
| Transport POIs | 0.443 | 0.875 | -1.823 | 4.191 | 6.014 | 0.020 | 0.208 |
| POI mixture | -0.033 | 0.063 | -0.234 | 0.159 | 0.393 | -0.069 | -0.031 |
| Road intersections | -0.043 | 0.206 | -0.907 | 0.657 | 1.565 | -0.134 | -0.056 |
| Bus stops | 0.051 | 0.168 | -0.660 | 1.096 | 1.755 | -0.022 | 0.035 |

location further from the city center. The spatial distributions of coefficients of local variables reveal the heterogeneous impact of the built environment (Figs 4 and 5).

For the home-based model, surprisingly, the city center is not significantly affected by most built environment factors (Fig 4). We do not conclude that the result violates the conclusions of current studies. Rather, it proves that the influence of the built environment is spatially heterogeneous. The residential population increases the commuting distance in eastern and southern suburbs, but it decreases commuting distance at the university town near the inner city. It has a significant influence on the Huangpu suburb in the east. Huangpu suburb has a strong jobs–housing connection with the city center, and a higher residential population is associated with longer commuting distance. However, at the university town, the residential population shortens the commuting distance. The provision of housing encourages employees in the university to live near their workplace, so jobs and housing are balanced. The work population reduces the commuting distance in suburban areas. Providing jobs locally would encourage people to work near their residence and reduce outgoing commuting in the southern suburbs, northern airport areas, and the eastern suburbs where job opportunities are not yet sufficient. Recreation facilities such as shopping, restaurants and public services have a positive impact in the northern airport suburbs and a negative impact in the southeastern suburbs. In the southeastern suburbs where there is a large residential community, better commercial development would encourage people to work near their residential neighborhoods. In contrast, in northern airport areas, recreation facilities increase home-based commuting distance

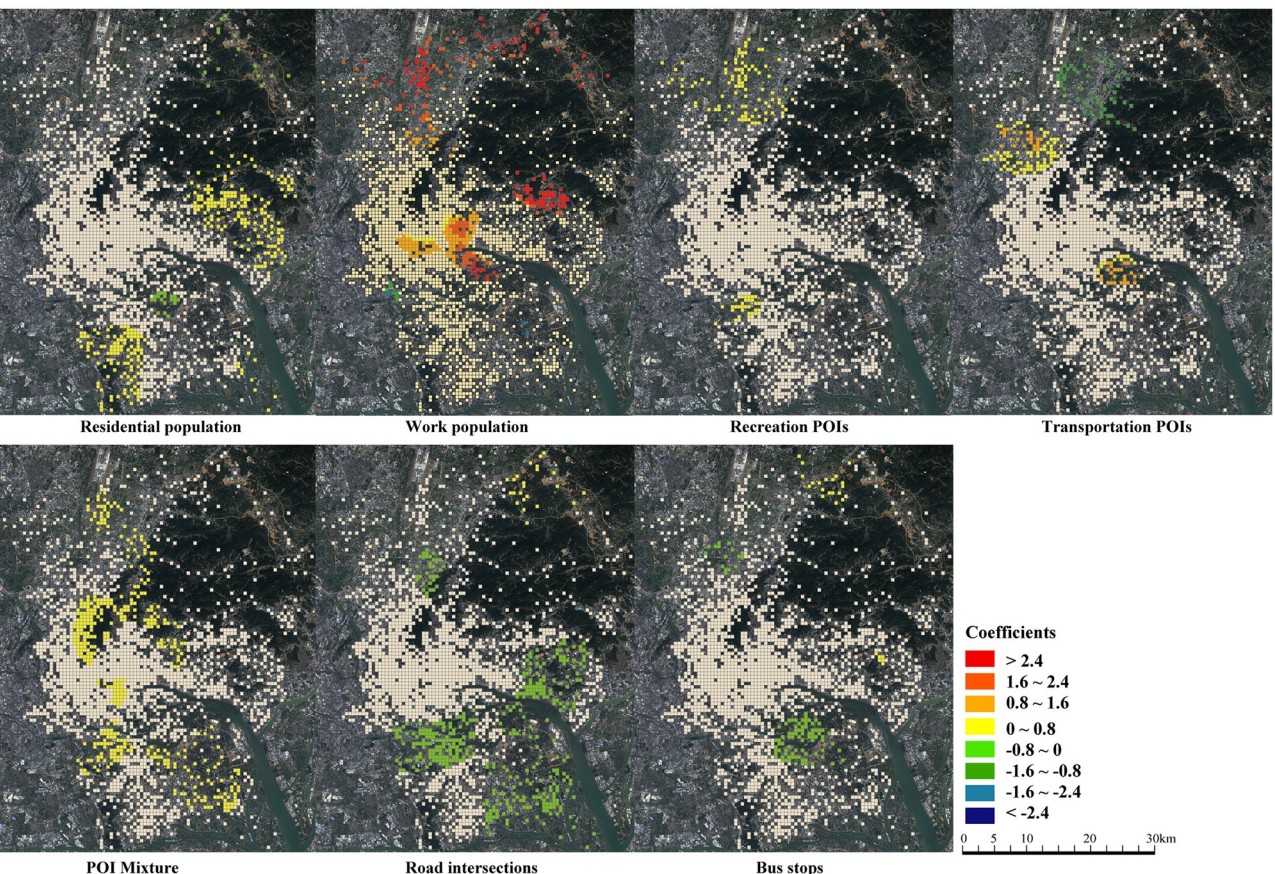

**Fig 4. Distributions of coefficients in the home-based model.**

slightly. The airport is a local administrative center. Well-developed public services would attract people to reside nearby, although they may work at other places. Transport facilities increase the home-based commuting distance in the northwestern wholesale market areas, where the provision of parking and car services encourages people to travel longer in the morning peak. POI mixture increases the commuting distance, which is contrary to previous study results. Traditionally it is believed that mixed urban function reduces commuting distance since different urban functions are located together [3]. We argue that Chinese cities have a different spatial context from North American cities which have experienced decentralization and suburbanization. Guangzhou has developed with highly mixed land use. The mixture in urban function implies a convenient living environment and services. It attracts employees to live there, although their workplace may not be near their residence. The density of road intersections would shorten the commuting distance in suburban areas, while it has little impact in the central areas. In the suburban areas where the road network is not as dense as the central areas, improving the design of the road network helps commuters choose closer workplaces. The number of bus stops also shortens the commuting distance in the southeast suburban areas, where there is the new developing university town in particular. Improving public transit would encourage people to work near their residential areas.

Different from the home-based model, population factors have a significant impact in the city center in the work-based model (Fig 5). The number of residents reduces the work-based

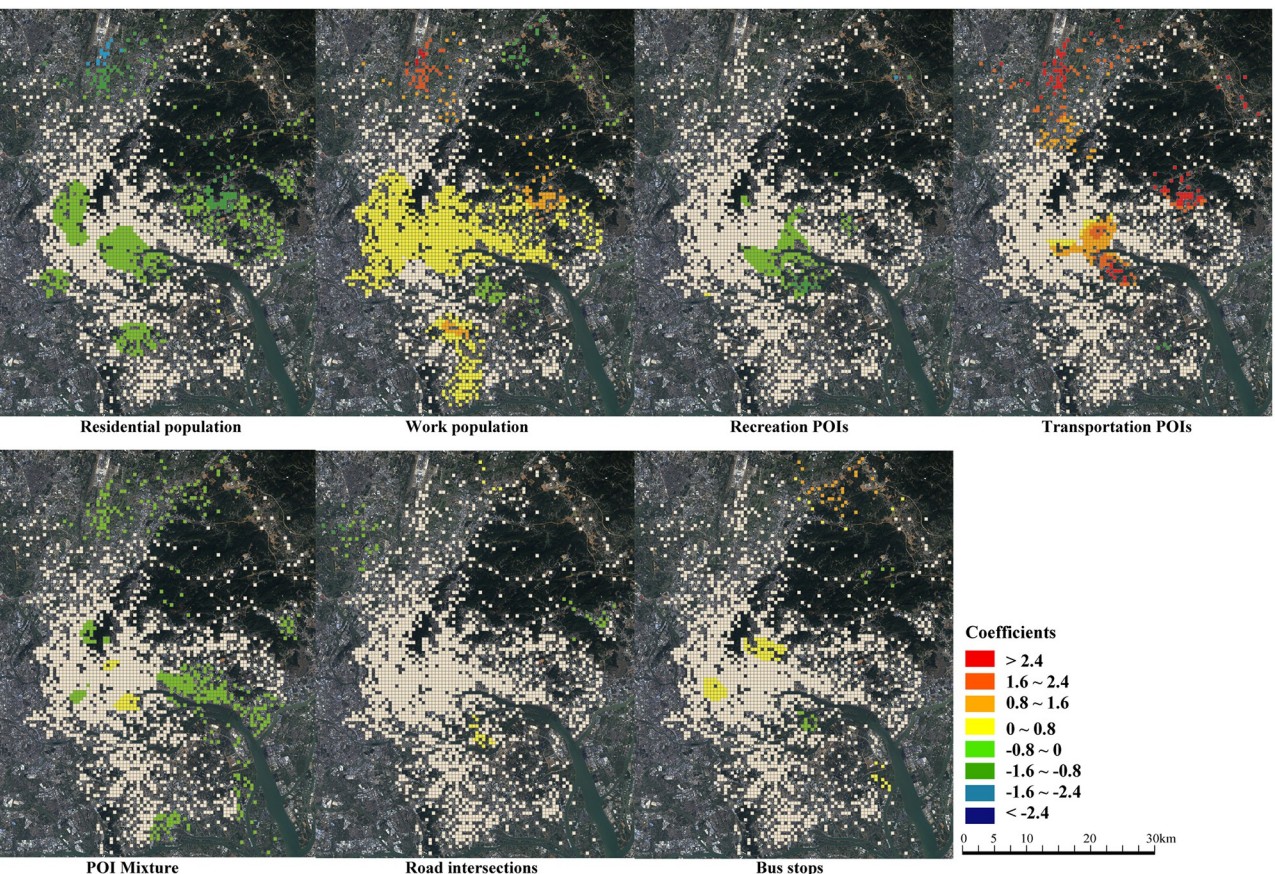

**Fig 5. Distributions of coefficients in the work-based model.**

commuting distance. Jobs are highly concentrated in the city center and sub-centers. Increasing residents in these areas would significantly decrease work-based commuting. The number of working people, in contrast, increases the work-based commuting distance. These results are consistent with previous findings that an imbalance between jobs and housing is associated with longer commuting [22,50]. Interestingly, the work population reduces the commuting distance in the newly developed university town. Providing more job opportunities would improve the jobs–housing balance, reducing work-based commuting distance. Recreation POIs negatively affect the commuting distance at the business center. Recreation facilities provide a convenient residential environment to attract people to live in. Employees are willing to live near their workplaces to get better commercial services. Transport facilities increase work-based commuting distances at the business center, the northern airport suburb, and the eastern high-tech industrial park. Well-developed parking facilities and car services provide good services for private cars, which makes it convenient for commuters to travel from distant locations. In contrast to the home-based model, POI mixture generally reduces work-based commuting distance. In suburban areas, mixed functions attract employees to live near their workplaces. The number of bus stops slightly increases the work-based commuting distances at the old town center. Good public transport services make it convenient for commuters to travel from distant locations.

**Table 4. BIC of two-step cluster models.**

| Number of Clusters | BIC | BIC Change | Ratio of BIC Changes | Ratio of Distance Measures |
|---|---|---|---|---|
| 1 | 25524.060 | | | |
| 2 | 22376.430 | -3147.630 | 1.000 | 1.356 |
| 3 | 20107.896 | -2268.534 | .721 | 1.879 |
| 4 | 18993.630 | -1114.265 | .354 | 1.087 |
| 5 | 17984.426 | -1009.204 | .321 | 1.060 |
| 6 | 17043.600 | -940.827 | .299 | 1.196 |
| 7 | 16289.617 | -753.983 | .240 | 1.221 |
| 8 | 15707.931 | -581.686 | .185 | 1.245 |
| 9 | 15279.649 | -428.283 | .136 | 1.052 |
| 10 | 14882.382 | -397.267 | .126 | 1.081 |

## Guidance for zonal planning policies

The study is significant in practice. Planners might propose less effective spatial development strategies when they are unaware of heterogeneous effects. To solve the problem of long commuting distance related to spatial heterogeneity, it is necessary to implement zonal strategies rather than a spatially homogeneous or city-wide strategy. A key question is what built environment factors significantly influence commuting at which locations. Uncovering the spatially heterogeneous relationship between the built environment and commuting distance would provide guidance.

The results from the GWR models have implicitly detected a double-edged effect of the built environment, such that the impact of a particular variable on home-based and work-based commuting can vary for a spatial unit. For example, the work population reduces home-based commuting distance but increases work-based commuting distance in suburban areas. The impact is also spatially heterogeneous. A key question is what the combined effects of all built environment variables are at a particular location. A two-step cluster model is applied to examine the double-edged effect of the built environment. It further answers the research question 3.

It is necessary to first decide an appropriate number of clusters. The model generates a series of clusters with up to 10 clusters (Table 4). Bayesian information criterion (BIC) is used to find the appropriate number of clusters. The greatest change between the two closest clusters indicates the most appropriate value. For the classification of three clusters, the BIC change of -2268.534 (0.721 ratio of change) is regarded as the greatest change. Accordingly, the spatial cells are classified into 3 clusters (Table 5).

Table 6 shows the two-step cluster result. Clusters 1–3 are visualized in Fig 6. Cluster 1 represents the central areas. Work population causes a double-edged effect with reduced home-based commuting distance but increased work-based commuting distance. The finding challenges traditional understanding of the commuting and built environment relationship. It also generates a challenge for urban planners in that a policy of jobs–housing balance may have an unintended effect. A good policy should balance outgoing and incoming commuting flows. POI mixture also has a double-edged effect. Mixed development is a good strategy for reducing the work-based commuting distance, yet not for the home-based commuting distance. Transport facilities and bus stops are significantly associated with longer work-based commuting distance. In job-rich areas, in particular, well-developed parking services and public transit services provide better accessibility for employees so that more commuters from distant locations work in these areas. It confirms the work-attractive nature of the transport facilities and public transit. Residential population and recreation POIs are associated with reduced work-based

**Table 5. Number of predicted variables by clusters.**

| Cluster | Resi_H | | | Work_H | | | Recr_H | | | Trans_H | | | Mix_H | | | Bus_H | | | Inter_H | | |
|---|---|---|---|---|---|---|---|---|---|---|---|---|---|---|---|---|---|---|---|---|---|
| | - | not sig. | + | - | not sig. | + | - | not sig. | + | - | not sig. | + | - | not sig. | + | - | not sig. | + | - | not sig. | + |
| 1 | 0 | 728 | 0 | 23 | 705 | | 0 | 728 | 0 | 0 | 728 | 0 | 0 | 670 | 58 | 0 | 728 | 0 | 57 | 671 | 0 |
| 2 | 14 | 366 | 176 | 504 | 52 | | 3 | 447 | 106 | 67 | 475 | 14 | 2 | 473 | 81 | 12 | 519 | 25 | 51 | 485 | 20 |
| 3 | 44 | 692 | 29 | 159 | 606 | | 54 | 711 | 0 | 0 | 679 | 86 | 0 | 634 | 131 | 85 | 680 | 0 | 296 | 466 | 3 |
| Total | 58 | 1786 | 205 | 686 | 1363 | | 57 | 1886 | 106 | 67 | 1882 | 100 | 2 | 1777 | 270 | 97 | 1927 | 25 | 404 | 1622 | 23 |

| Cluster | Resi_W | | | Work_W | | | Recr_W | | Trans_W | | | Mix_W | | | Bus_W | | | Inter_W | | |
|---|---|---|---|---|---|---|---|---|---|---|---|---|---|---|---|---|---|---|---|---|
| | - | not sig. | + | - | not sig. | + | - | not sig. | - | not sig. | + | - | not sig. | + | - | not sig. | + | - | not sig. | + |
| 1 | 292 | 436 | 0 | 0 | 0 | 728 | 112 | 616 | 0 | 579 | 149 | 119 | 576 | 33 | 0 | 643 | 85 | 0 | 728 | 0 |
| 2 | 317 | 239 | 0 | 24 | 182 | 350 | 4 | 552 | 5 | 380 | 171 | 163 | 393 | 0 | 7 | 513 | 36 | 21 | 528 | 7 |
| 3 | 0 | 762 | 3 | 67 | 688 | 10 | 43 | 722 | 6 | 716 | 43 | 72 | 693 | 0 | 11 | 742 | 12 | 20 | 730 | 15 |
| Total | 609 | 1437 | 3 | 91 | 870 | 1088 | 159 | 1890 | 11 | 1675 | 363 | 354 | 1662 | 33 | 18 | 1898 | 133 | 41 | 1986 | 22 |

Abbreviations.

Home-based model: Resi_H (residential population), Work_H (work population), Recr_H (recreation POIs), Trans_H (transport POIs), Mix_H (POI mixture), Bus_H (bus stops), Inter_H (road intersections).

Work-based model: Resi_W (residential population), Work_W (work population), Recr_W (recreation POIs), Trans_W (transport POIs), Mix_W (POI mixture), Bus_W (bus stops), Inter_W (road intersections).

commuting distance. For local employees, they are encouraged to live near their workplace because of the good living environment and convenient commercial and public services.

Cluster 2 represents the outer suburbs. The residential population, work population, transport facilities and POI mixture have double-edged effects on commuting distance. Residential population increases home-based commuting distance and reduces work-based commuting, while work population reduces the home-based commuting and increases the work-based commuting. The double-edged effect of POI mixture in the outer suburbs is similar to the central areas. While transport facilities improve home-based commuting distance, they worsen work-based commuting. Recreation facilities also increase home-based commuting distance. Better commercial and public service development attracts employees to live in an area. For non-local employees, their workplaces are mismatched with their residences so their home-based commuting distances are increased.

Most cells in Cluster 3 are inner suburbs. Several factors have a negative impact on the commuting distance in Cluster 3. Work population and recreation facilities shorten both home-based and work-based commuting distances. The road network density and the number of bus stops are associated with shorter home-based commuting distance. Transport facilities increase both home-based and work-based commuting distances.

The results generate a further question about how to develop zonal planning policies according to the complex relations between the built environment and commuting (question 4). Based on the above analysis, some guidance is provided. According to the two-step model, the built environment's spatially heterogeneous impact can be clustered into three types of

**Table 6. Cluster description.**

| Cluster | Name | N | % | Variables with a positive impact | Variables with a negative impact |
|---|---|---|---|---|---|
| 1 | Central areas | 728 | 35.50% | Mix_H, Work_W, Trans_W, Bus_W | Work_H, Inter_H, Resi_W, Recr_W, Mix_W |
| 2 | Outer suburbs | 556 | 27.10% | Resi_H, Recr_H, Mix_H, Work_W, Trans_W | Work_H, Trans_H, Resi_W, Mix_W |
| 3 | Inner suburbs | 765 | 37.30% | Trans_H, Mix_H, Trans_W | Work_H, Recr_H, Bus_H, Inter_H, Work_W, Recr_W, Mix_W |

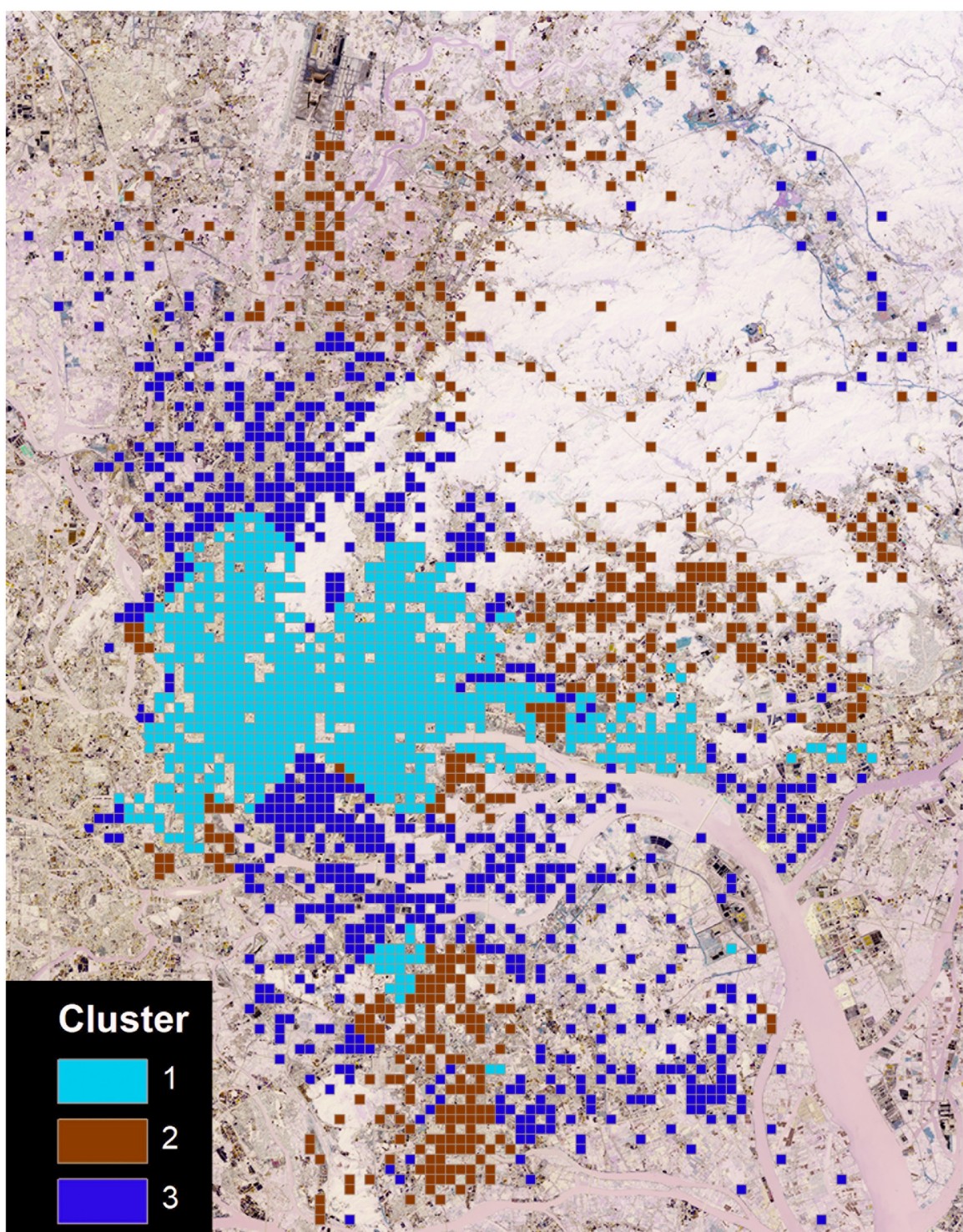

**Fig 6. The spatial distribution of clusters.**

areas: central areas, inner suburbs, and outer suburbs. The government should develop different spatial planning strategies for the three types of areas. First, the government should be aware of the double-edged effect of a 'diversity' strategy. Mixed development is a good strategy for reducing work-based commuting distance. However, it does not help solve the problem of long home-based commuting. In inner suburbs and outer suburbs which have long home-based commuting distance, mixed development should not be emphasized. Second, in the central areas, housing and recreation should be provided in job-rich areas. Residential population is associated with shorter home-based and work-based commuting distances. Recreation facilities also help reduce work-based commuting distance. However, transport facilities and public transit increase the work-based commuting distance. The provision of transport facilities and public transit should be increased outside the central areas and spatially equally across the whole city. Third, increasing the work population, recreation facilities, public transit services and road network density helps improve the long commuting problem in inner suburbs. Fourth, residential population, work population, transport facilities and mixed development have a double-edged effect on the commuting distance in the outer suburbs. Since the problem in outer suburbs is the long home-based commuting distance not the work-based distance, we suggest that more jobs and transport facilities should be provided in the outer suburbs to reduce home-based commuting distance.

## Conclusions and discussions

This study investigated the spatially heterogeneous impact of the built environment on commuting distance using a massive mobile phone dataset in Guangzhou city, China. The travel distance of commuters was found to follow an exponential distribution. Geographically weighted regression models were applied to investigate the spatial heterogeneous impact of built environment variables on distance decay parameters.

Results showed that the impact of the built environment on commuting distance is spatially heterogeneous. The result can guide zonal planning policies. Based on a two-step cluster model, the urban space is classified into three clusters of central areas, inner suburbs and outer suburbs. Results revealed that the built environment has a double-edged effect on commuting distance, differentiated by home-based and work-based commuting. Residential population, recreation facilities, and mixed development are residence-attractive factors. In general, they have a positive impact on home-based commuting distance and a negative impact on work-based commuting distance. The work population and transport facilities are work-attractive factors. Their impact on commuting distance is contrary to the residence-attractive factors such that home-based commuting distances are decreased and work-based commuting distances are increased.

These findings have provided a new understanding of the relationship between the built environment and commuting distance. The relationship is dominated by different mechanisms—the market mechanism and the individual choice mechanism. From the results that the relationship is spatially heterogeneous, we can see the encounter between the two mechanisms across the urban space. The market mechanism dominates the relationship at the business center. For the home-based commuting at the business center, most built environment factors' influence is not significant. It challenges the co-location theory, which believes that the co-location of jobs and houses would shorten the commuting distance [2]. Rather, our result supports the opponent opinion from Giuliano [6] that the co-location of jobs and housing is not significantly associated with the shorter commuting distance. The reason is that the dominant mechanism at the business center is not individual choice but the market. Individuals' bidding ability is weaker than industrial firms, and they cannot freely choose their residential

locations in the city center. Thus, they choose to live further and abandon the need of saving commuting time. The market mechanism works on the work-based commuting distance. The work-based commuting distance reflects how far the workplace can attract people from. Because of the industrial agglomeration, the business center attracts workers from the whole city. Also, oversupplying industrial land reduces the housing land, and workers are forced to reside further. These reasons lead to the long work-based commuting distance [51].

Interestingly, the individual choice mechanism dominates the relationship between the built environment and the commuting distance again at the residential areas. From the home-based perspective, our results are consistent with other home-based studies [17–21]. However, it is surprising to see that mixed development is not always a good strategy for reducing commuting distance as previous studies have found [33–35]. In this study, mixed development is associated with longer home-based commuting trips. A mixed development strategy should be emphasized more in job-rich areas than in housing-rich areas. Our explanation is that the mixed-development strategy [33,34] is also based on the individual choice mechanism. In residential areas, mixed development provides better public and commercial services. When people have free choices on residential locations, they prefer mix developed areas for better public service and convenient living environment. However, they only have free choices of housing locations in residential areas where the market mechanism has little influence. Since individuals' preferred housing locations are outside the business center, they are far away from their workplaces. As a result, land-use mixture increases the home-based commuting distance.

From the literature, we can see that the relationship between the built environment and the commuting distance is different from study to study. This research provides a deep insight into these different opinions. In summary, because of shift between the market mechanism and individual choice mechanism, the theory to explain the built environment and the commuting distance relationship is not unique across the whole urban space. Rather, the relationship is spatially heterogeneous. The individual choice mechanism and the co-location theory are applicable for home-based studies, and work-based studies should apply the market mechanism. This finding contributes to the theory of the built environment and travel relationship that theoretical assumptions have different application conditions.

The study is limited in not incorporating individual socioeconomic attributes. Socioeconomic attributes are important factors which affect people's commuting behavior. It is a common approach to explore the influence of people's socioeconomic characteristic on the commuting distance, particularly in disaggregate analysis [17]. In this study, we did not consider socioeconomic attributes as independent variables. Our argument is that government can implement spatial planning measures to decrease the commuting distance by improving the built environment, but socioeconomic attributes cannot be easily changed and are not effective policy measures for the government. Nevertheless, socioeconomic attributes still have potential influence on commuting behavior. Excluding them would cause biased results of the built environment and the commuting distance relationship. Realizing the shortcoming, we will incorporate individual-level data to further explore the behavioral drivers of commuting distance in future research.

## Supporting information

**S1 Data.**
(XLSX)

## Author Contributions

**Conceptualization:** Zhong Zheng, Suhong Zhou.

**Data curation:** Zhong Zheng, Xingdong Deng.

**Formal analysis:** Zhong Zheng.

**Funding acquisition:** Zhong Zheng, Suhong Zhou.

**Investigation:** Zhong Zheng.

**Methodology:** Zhong Zheng, Suhong Zhou.

**Project administration:** Suhong Zhou, Xingdong Deng.

**Resources:** Xingdong Deng.

**Software:** Zhong Zheng.

**Supervision:** Suhong Zhou, Xingdong Deng.

**Validation:** Zhong Zheng, Suhong Zhou.

**Visualization:** Zhong Zheng.

**Writing – original draft:** Zhong Zheng.

**Writing – review & editing:** Suhong Zhou.

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
