## [Decision Letter · Decision Letter 0]

21 Sep 2020

PONE-D-20-19283

Exploring spatially heterogeneous impact of built environments: The double-edge effect on home-based and work-based commuting

PLOS ONE

Dear Dr. Zhou,

Thank you for submitting your manuscript to PLOS ONE. After careful consideration, we feel that it has merit but does not fully meet PLOS ONE’s publication criteria as it currently stands. Therefore, we invite you to submit a revised version of the manuscript that addresses the points raised during the review process.

We look forward to receiving your revised manuscript.

Kind regards,

Wenjia Zhang

Academic Editor

PLOS ONE

Journal Requirements:

2.We note that you have indicated that data from this study are available upon request. PLOS only allows data to be available upon request if there are legal or ethical restrictions on sharing data publicly. For more information on unacceptable data access restrictions, please see http://journals.plos.org/plosone/s/data-availability#loc-unacceptable-data-access-restrictions.

4. We note you have included a table to which you do not refer in the text of your manuscript. Please ensure that you refer to Table 6 in your text; if accepted, production will need this reference to link the reader to the Table.

5.We note that the figures in your submission contain [map/satellite] images which may be copyrighted. All PLOS content is published under the Creative Commons Attribution License (CC BY 4.0), which means that the manuscript, images, and Supporting Information files will be freely available online, and any third party is permitted to access, download, copy, distribute, and use these materials in any way, even commercially, with proper attribution. For these reasons, we cannot publish previously copyrighted maps or satellite images created using proprietary data, such as Google software (Google Maps, Street View, and Earth). For more information, see our copyright guidelines: http://journals.plos.org/plosone/s/licenses-and-copyright.

1.    You may seek permission from the original copyright holder of the figures to publish the content specifically under the CC BY 4.0 license. 

Additional Editor Comments (if provided):

Please carefully address each of the comments by the two reviewers, particularly clarify your literature and contribution.

Reviewers' comments:

Reviewer's Responses to Questions

**Comments to the Author**

1. Is the manuscript technically sound, and do the data support the conclusions?

Reviewer #1: Partly

Reviewer #2: Partly

2. Has the statistical analysis been performed appropriately and rigorously? 

Reviewer #1: Yes

Reviewer #2: Yes

3. Have the authors made all data underlying the findings in their manuscript fully available?

Reviewer #1: No

Reviewer #2: No

4. Is the manuscript presented in an intelligible fashion and written in standard English?

Reviewer #1: Yes

Reviewer #2: Yes

5. Review Comments to the Author

Reviewer #1: This paper aims to examine the spatially heterogeneous impact of built environment on commuting distances using mobile phone data in Guangzhou, China. Some questions need to be addressed before consideration of publication:

1. The Literature review section needs to be reorganized. The two subsections have both reviewed the impacts of built environment on commuting. The spatial heterogeneity is not clear in the review section.

2. What is the difference between home-based and work-based commuting? Please make it clearer. In many cases, they might be similar. Commuting is influenced by built environment at both residential location and workplace.

3. Some figures are in poor quality, please modify them. Moreover, it had better provide some figures to illustrate the spatial distribution of built environment in Guangzhou before running the model.

4. Please give more details on the data.

5. There are several grammar errors in the paper. Please improve the language.

Reviewer #2: The paper uses geographically weighted regression models to examine the spatially heterogeneous effect of the built environment on commute. The authors concluded that the impact from built environment might vary over space and also be different for home-based and work-based commuting. I have several major concerns/suggestions:

• The contribution of the paper is weak. The authors need to highlight the values of your paper. If the value is to examine the spatial heterogeneity in the relationships, literature review needs to be rewritten with the focus of spatial context. What are the study areas of the existing studies? Is it region-wide, city-wide, or country-wide study? Could the complicating results be due to differences in spatial context? Also, need to add a conceptual framework explaining why the relationships could be spatially varied. Why separating home-based and work-based commutes and why the effects could be double-edged? Your conceptual framework should built upon existing studies.

• “Dependent variable is decay parameter beta in Eq.1. It represents the mean of the distance distribution.” Is it the mean of the distance or the slope of the decay? Please explain why using beta as dependent variable and refine your interpretation of beta. If it represents the mean, why not simply using the mean of all local distances? Your result interpretation need to be consistent with your definition of beta. For example, on page 9, “Residential population increases commuting distance….” It should be residential population increases beta, correct?

• The independent variables need clarifications. How are independent variables selected? For instance, why commercial and recreation function could influence commute? Are the built environment variables measured for a single cell? Are surrounding cells included? “Infrastructure POIs represent the transportation facilities”. What types of transportation facilities? Why Closeness and Distance to center are global variables?

• The current study area is the inner city of Guangzhou? Any particular reasons why the paper only focus on the inner city?

• Please also add more discussions about the value of the cluster analysis. Within a single cluster, a variable could also have double-edged effect on commute, right?

Other editorial suggestions:

• On page 2, “Most studies of commuting and built environment are based on global models, which assume the impact of built environment is spatially unique.” It should be “spatially homogeneous”.

• “Human activity in a city is naturally heterogeneous, i.e. the distributions of populations, daily activities and traffic flows.” the writing needs to be revised to focus on the heterogeneity of the relationships (instead of human activity).

• On page 3, “[13] investigated the relation of VMT [vehicle miles traveled] and jobs-housing ratio.” Double check the citation requirement of the journal. Normally, the paper still needs to list the authors’ names in citations.

• On page 5, “to reveal the heterogeneous effect explicitly, it is necessary to apply a local model, a geographically weighted regression model e.g., to give an overview of the structural relation between commuting and built environment of the whole city”. Move “e.g.” before “a geographically weighted regression model”.

6. PLOS authors have the option to publish the peer review history of their article (what does this mean?). If published, this will include your full peer review and any attached files.

Reviewer #1: No

Reviewer #2: No

---

## [Author Response · Author response to Decision Letter 0]

28 Dec 2020

GENERAL COMMENTS TO THE EDITOR AND REVIEWERS:

We are pleased to resubmit our revised manuscript of PONE-D-20-19283, which entitles ‘Spatially heterogeneous and double-edged effect of built environments on aggregated home-based and work-based commuting’. We appreciate thoughtful comments and constructive suggestions of the reviewers. 

We explain how we revised the paper based on each comment as outlined below. 

RESPONSE TO REVIEWER #1:

This paper aims to examine the spatially heterogeneous impact of built environment on commuting distances using mobile phone data in Guangzhou, China. Some questions need to be addressed before consideration of publication:

1. The Literature review section needs to be reorganized. The two subsections have both reviewed the impacts of built environment on commuting. The spatial heterogeneity is not clear in the review section.

Response: Thanks for your comments. We totally rewrote the literature review part (lines 129-235). The new one has reviewed three aspects of spatial heterogeneity analysis: differences between centers and suburbs, a multi-level model, and a GWR model. It enhances the theoretical construction of this study according to your suggestion. 

2. What is the difference between home-based and work-based commuting? Please make it clearer. In many cases, they might be similar. Commuting is influenced by built environment at both residential location and workplace.

Response: Thanks for your suggestion. Home-based and work-based commuting are differentiated by two groups of commuters – residents and employees - in the same spatial unit. They are not differentiated by land-use of residence and work. For example, departure trips in the morning in a work-rich area is home-based commuting because we are observing commuters who live there. A same built environment variable’s impact is on two groups of people, that’s why we separate them and propose the ‘double-edged effect’ hypothesis. We gave a more details that:

‘It is important to note that, for the same spatial unit or neighborhood, there might be two distinct groups of commuters with totally different behavior: residents and employees. The former is home-based commuters who depart from the spatial unit, while the latter is work-based commuters who arrive at the spatial unit. Because of the behavioral differences of two commuter groups, it is necessary to differentiate between home-based and work-based commuting. ‘(lines 79-85). 

‘In this study, an underlying hypothesis is that the built environment’s impact on home-based and work-based commuting may be different or even contrary, causing the double-edged effect. To examine the double-edged effect, it is necessary to analyze the relation of commuting and built environment from both home-based and work-based perspectives.’ (lines 91-96)

3. Some figures are in poor quality, please modify them. Moreover, it had better provide some figures to illustrate the spatial distribution of built environment in Guangzhou before running the model.

Response: Yes, we have improved quality of figures. The poor figure quality may be due to the compression of figures in the submission system. Original figures can be downloaded from the link at right upper corner of figure pages. We also provide the spatial distribution of built environment variables in Fig 2. 

4. Please give more details on the data.

Response: We gave more details on the cellphone data and POI data. For example, we added in lines 254-260 that ‘The cellphone data have identified 13.7 million commuting trips in one month.’ And that ‘Points of interest (POIs) data are also applied to represent the built environment. A piece of ‘point of interest’ data records information of a coordinate location and a functional type of a spatial facility from a navigation map. POIs data are sourced from the Baidu map and provided by the Daodaotong company. Totally 27349 pieces of POIs are used in this study.’

We also give details of recreation and transport POIs in variable description part (lines 312-315) that ‘Recreation POIs include dining (20.6%), public service (15.7%), entertainment (11.5%) and shopping (52.2%). Transport POIs include the parking logs (51.6%), car services (34.0%), and important transport navigation spots such as toll gates, bridges, train/bus stations, etc. (14.4%).’ 

5. There are several grammar errors in the paper. Please improve the language.

Response: Yes, we did another round of proofreading throughout the article. We have revised more than 200 grammar errors or inappropriate sentences. 

RESPONSE TO REVIEWER #2:

The paper uses geographically weighted regression models to examine the spatially heterogeneous effect of the built environment on commute. The authors concluded that the impact from built environment might vary over space and also be different for home-based and work-based commuting. I have several major concerns/suggestions:

• The contribution of the paper is weak. The authors need to highlight the values of your paper. If the value is to examine the spatial heterogeneity in the relationships, literature review needs to be rewritten with the focus of spatial context. What are the study areas of the existing studies? Is it region-wide, city-wide, or country-wide study? Could the complicating results be due to differences in spatial context? Also, need to add a conceptual framework explaining why the relationships could be spatially varied. Why separating home-based and work-based commutes and why the effects could be double-edged? Your conceptual framework should built upon existing studies.

Response: Thanks for your thoughtful suggestions. In this round of edit, we made a major revision on the literature review part. We totally rewrote it and emphasized the spatial heterogeneity. Related studies are reviewed by three aspects: differences between centers and suburbs, a multi-level model, and a GWR model. See contexts in lines 130-235.

Commuting is basically a city-wide issue, although some studies compare different cities’ commuting and built environment relations. This study narrows it in a city-wide spatial context. We state it in contexts ‘there are still controversies on how they affect commuting in a city-wide spatial context (lines 46-47)’ and ‘taking a city or a part of a city as cases, studies of the heterogeneous impact of the built environment can be summarized in three aspects. (lines 140-142).’ Exploring the spatial heterogeneity in a city-wide improves our understanding of the urban space in a finer scale that ‘Since commuting is generally a city-wide issue, the heterogeneous effect should be revealed explicitly on a finer spatial scale. Therefore, it is necessary to apply a local model, a geographically weighted regression model e.g., to give an overview of the structural relation between commuting and the built environment of the whole city. (lines 230-233)’

Separating home-based and work-based commuting is because they represent different behavior of two commuter groups, those depart from a spatial unit and those arrive at a spatial unit. A same built environment variable’s impact is on two groups of people, that’s why we separate them and propose the ‘double-edged effect’ hypothesis. We gave more details:

It is important to note that, for the same spatial unit or neighborhood, there might be two distinct groups of commuters with totally different behavior: residents and employees. The former is home-based commuters who depart from the spatial unit, while the latter is work-based commuters who arrive at the spatial unit. Because of the behavioral differences of two commuter groups, it is necessary to differentiate between home-based and work-based commuting. (lines 79-85). 

We also added a conceptual framework in Fig 1, and the article is re-organized by answering four research questions according to the framework.

• “Dependent variable is decay parameter beta in Eq.1. It represents the mean of the distance distribution.” Is it the mean of the distance or the slope of the decay? Please explain why using beta as dependent variable and refine your interpretation of beta. If it represents the mean, why not simply using the mean of all local distances? Your result interpretation need to be consistent with your definition of beta. For example, on page 9, “Residential population increases commuting distance….” It should be residential population increases beta, correct?

Response: Thanks for your comment. Please see our response as follows: ‘The decay parameter also represents the expectation of the distance distribution in an exponential distribution. is not simply a mean of commuting distances. Rather, we should treat it as a weighted average of that, according to the property of an exponential distribution. In other words, the expectation value also shapes the slope of the curve. It represents either a high probability of longer distances and a low probability of shorter distances (large ), or a high probability of shorter distances and a low probability of longer distances (small ). The decay parameters of commuting distance distributions are differentiated by departure trips and arrival trips, which represent home-based and work-based commuting respectively. (lines 283-291)’

• The independent variables need clarifications. How are independent variables selected? For instance, why commercial and recreation function could influence commute? Are the built environment variables measured for a single cell? Are surrounding cells included? “Infrastructure POIs represent the transportation facilities”. What types of transportation facilities? Why Closeness and Distance to center are global variables?

Response: Thanks for your questions. We answered them in our manuscript: Independent variables (Table 1) are selected according to ‘Ds’ measurement [42] of the built environment, including density, diversity, design, destination accessibility, etc. (lines 292-293)

Influences from commercial and recreation function on commuting can refer to previous studies. They are explained in lines 304-309.

All variables are measured in a single cell without including surrounding cells. Since the GWR model itself is a spatially weighted algorithm considering the built environment’s impact from surrounding spatial units, there is no need to measure the built environment from adjacent cells again. (see lines 343-347)

Transport POIs include the parking logs (51.6%), car services (34.0%), and important transport navigation spots such as toll gates, bridges, train/bus stations, etc. (14.4%). Transport facilities are mainly associated with vehicle trips. (see lines 314-315)

Built environment variables should be further defined as local ones and global ones in a GWR Model. A local variable means its coefficient value varies across different locations, while a global variable has a unique coefficient value across the entire study area like a linear regression model. Variables not capable to capture the spatial relation, such as residential population, work population, recreation POIs, transportation POIs, and POI mixture, are local variables. The value of local variables varies across different locations. Variables of closeness and distance to the center, differently, are global variables since they represent the location effect. These variables themselves can describe spatial relation to the city center. (lines 335-343)

• The current study area is the inner city of Guangzhou? Any particular reasons why the paper only focus on the inner city?

Response: Actually the study area is the urbanized area of Guangzhou. We explained it in texts that ‘the study area is the urbanized area but not the whole city. The urbanized area of the city has an area of 2435.7 km2 and a population of 11.5 million in 2019[ Guangzhou Statistics Bureau: http://tjj.gz.gov.cn/tjdt/content/post_5727607.html]. The city’s population is concentrated in the urbanized area that 75% of the population is distributed at 32% area. The rest un-urbanized area is mainly rural and forest land with sparsely distributed residential settlements. That is the reason why we select the urbanized area as the study case. (lines 239-244)’

• Please also add more discussions about the value of the cluster analysis. Within a single cluster, a variable could also have double-edged effect on commute, right?

Response: Yes, based on a new conceptual framework, the value of the cluster analysis is stated in lines 452-460: ‘The results from GWR models have implicitly detected a double-edged effect of the built environment, that the impact of a particular variable on home-based and work-based commuting could be contrary at a spatial unit. For example, the work population decreases home-based commuting distances but increases work-based commuting distances in suburban areas. The impact is also spatially heterogeneous. A key question is what the combined effects of all built environment variables are at a particular location. To reveal the double-edged effect explicitly, a two-step cluster model is applied to further examine the area-based impact of the built environment and answer Research Question 3. ‘

Other editorial suggestions:

• On page 2, “Most studies of commuting and built environment are based on global models, which assume the impact of built environment is spatially unique.” It should be “spatially homogeneous”.

Response: Yes, we revised it. 

• “Human activity in a city is naturally heterogeneous, i.e. the distributions of populations, daily activities and traffic flows.” the writing needs to be revised to focus on the heterogeneity of the relationships (instead of human activity).

Response: Yes, we revised it as ‘However, their relations in a city are naturally heterogeneous, causing spatial issues [7] because of locational effects.’

• On page 3, “[13] investigated the relation of VMT [vehicle miles traveled] and jobs-housing ratio.” Double check the citation requirement of the journal. Normally, the paper still needs to list the authors’ names in citations.

Response: We have revised them according to the citation format requirement. 

• On page 5, “to reveal the heterogeneous effect explicitly, it is necessary to apply a local model, a geographically weighted regression model e.g., to give an overview of the structural relation between commuting and built environment of the whole city”. Move “e.g.” before “a geographically weighted regression model”.

Response: Yes we moved it.

---

## [Decision Letter · Decision Letter 1]

7 Apr 2021

PONE-D-20-19283R1

Spatially heterogeneous and double-edged effect of built environments on aggregated home-based and work-based commuting

PLOS ONE

Dear Dr. Zhou,

Thank you for submitting your manuscript to PLOS ONE. After careful consideration, we feel that it has merit but does not fully meet PLOS ONE’s publication criteria as it currently stands. Therefore, we invite you to submit a revised version of the manuscript that addresses the points raised during the review process.

We look forward to receiving your revised manuscript.

Kind regards,

Wenjia Zhang

Academic Editor

PLOS ONE

Additional Editor Comments (if provided):

Thanks for submitting your manuscript to PloS One. Because a previous reviewer declined to review the paper, we had to find another reviewers to read your manuscript. It takes some time and I have now heard back from two reviewers, who are still not happy with the manuscript, although the manuscript is significantly improved. I will solicit once more the advice of the reviewer, and if the critical points are not resolved I will have to stop moving on due to the requirement of the journal. So please do carefully address each point of the comments.

Besides, I quickly went through your manuscript and I think you should address and clarify the following questions as well:

1. There are lots of language issues, still. For example, “rich literatures”, “commuting distances”, “built environment’s impact”, “considering…, conclusions”, Two “however in Line 57”, and so on. Please do find a native proofreading editor to go through the whole manuscript.

2. Similarly, some statements are vague and confusing. Like the paper title, which features of commuting do you focus on? Volume of commuters? Commuting means a lot, it could be distance, time, volume or more. Be specific.

In the abstract, “the analysis generally assumes…”, such a statement is not academically rigorous, because it is not a “general” truth in built environment-travel studies. “To fill in the research gaps…” what gaps? It needs to be specific. Also, what are “home-based” and “work-based perspective”? They may be explained in the manuscript, but it confuses the readers without much context in the Abstract. I cannot point out all the academic writing issues here, but you should rewrite some sentences in the whole manuscript to make each sentence clear and connected with context.

3. As mentioned by one reviewer, the definition of home-based and work-based commuters are weird. Each commuter is both a resident and an employee. So how can we say “residents and employees” are “two distinct groups of commuter” (Line 80)? Are they just similar to the concepts of Trip attraction and trip departure in traditional transport literature.

Now I see your definition of “commuting” as “decay parameters of probabilistic distributions of commuting distances” what is P,D in Equation (1), how beta is estimated is unclear, some examples to interpret beta could be helpful. Do you need to standardize the data since you use an estimated parameter to represent commuting. So another issue here is how to define beta as what kind of commuting feature in your study. You can’t generally say it as “commuting”, as pointed out in Comment #2. Why not defining clearly what beta means, then using the a very clear concept to replace the vague statement of “commuting”. If I don’t misunderstand your definition, you actually look at the probability of long-distance commuting departing versus arriving at a zone (spatial cell), right? Is that what the “home-based” versus “work-based” commuting represents?

4. Related to the conceptualization of heterogeneity, or the “contextual effect”, you may refer to Zhang and Zhang 2018a,b in JPER and Urban Studies.

Reviewers' comments:

Reviewer's Responses to Questions

**Comments to the Author**

1. If the authors have adequately addressed your comments raised in a previous round of review and you feel that this manuscript is now acceptable for publication, you may indicate that here to bypass the “Comments to the Author” section, enter your conflict of interest statement in the “Confidential to Editor” section, and submit your "Accept" recommendation.

Reviewer #3: (No Response)

Reviewer #4: (No Response)

2. Is the manuscript technically sound, and do the data support the conclusions?

Reviewer #3: Yes

Reviewer #4: (No Response)

3. Has the statistical analysis been performed appropriately and rigorously? 

Reviewer #3: I Don't Know

Reviewer #4: (No Response)

4. Have the authors made all data underlying the findings in their manuscript fully available?

Reviewer #3: (No Response)

Reviewer #4: (No Response)

5. Is the manuscript presented in an intelligible fashion and written in standard English?

Reviewer #3: (No Response)

Reviewer #4: (No Response)

6. Review Comments to the Author

Reviewer #3: 1.According to reponse" home-based commuters who depart from the spatial unit,

while the latter is work-based commuters who arrive at the spatial unit", Usually a person starts from space unit A(home) to space B(work) , there is a problem: The same travel behavior is not only the starting point of space a, but also the arrival point of space B, should it be counted twice as both home and work based commuting?Another problem is that he work and resident at same space unit.How to Identify and differentiated two groups of commuters – residents and employees through the data processing, please give processing flow and why did you do that.

2.The independent variables need clarifications. How are dependent variables indicators?

3.As the REVIEWER #2 suggest"Also, need to add a conceptual framework explaining why the relationships could be spatially varied". The figure added is so simple and cannont explain, more detailed influence action path and mechanism are needed.

Reviewer #4: This is an interesting study, and I think the authors' data material and their findings of differentiated relation between land use and commuting in 4 different clusters can potentially result in a published article of high importance. However, in its present form, the paper has some serious shortcomings that must be amended before I can recommend publication.

1. For one thing, the author uses two ambiguous concepts (i.e. home-based commuting, work-based commuting). Of course, I think I understand what the meaning is here; each spatial unit may become a residential area or workplace area, depending on the direction of commuting. But why not directly use the residential area and workplace area here, or you can define a spatial unit to have both residential area and workplace area attributes. In short, ‘home-based commuting’ and ‘work-based commuting’ are not general academic terms.

2. In Literature review section, the author needs to add some literature about the relation between built environment of residential area and workplace area, and the commuting. You can easily search for them on the google scholar. This is the most relevant research to your research. Further, you may find that your research question is not a huge gap.

3. In the framework, the author does not control any socioeconomic attributes, which may cause inaccurate results. For aggregated data, both statistical yearbooks and censuses can provide basic socioeconomic attributes in Chinese city, such as GDP, gender ratio, and elderly population ratio. Second, this research used a “5Ds” framework, but ignored Design and Distance to transit. Why? Because public transit accessibility, such as distance to bus/subway station or density of bus/subway station, is generally considered to be a very important factor affecting commuting. I think the author needs a very convincing reason here or needs to re-examine the framework.

4. For Variable Section, functional facility density belongs to Diversity rather than the Density (Pg. 15, line 295, 304-306; Ewing and Cervero 2010). Second, why choose Zhujiang New Town as the city center, and is Guangzhou a monocentric city? (Pg. 17, line 332-334). However, based on your statement: “The provision of housing encourages employees in the university to live near their workplaces, so the jobs-housing would be balanced. The work population decreases commuting distance in suburban areas (Pg. 22, line 413-415)”. In traditional monocentric city research, home-work distance is positively correlated with the distance to the city center. If you find that there is a low commuting distance area in the suburbs of Guangzhou, then I think Guangzhou may not be a monocentric city. So, you may need to re-examine how many city centers there are in Guangzhou. Besides, Closeness is not a general built environment variable; why use Closeness rather than road network distance to city center.

5. Method section lacks an introduction to Two-step cluster models.

6. I suggest that the author add an introduction map of the research area, may be in the Study area section, otherwise it is difficult for readers to understand where the airport suburb, university town, and Huangpu suburb are.

7. Pg. 3, line 55: the authors write: "Among these reasons, the spatial context is the most important one". Why? You need some evidence.

8. Pg. 5, line 86: the authors write: "Home-based commuting represents the morning traffic peak and work-based commuting represents the evening traffic peak". I don’t agree, because commuting is two-way. Both of residential area and workplace area are very congested in the morning and evening.

9. Pg. 5, line 98: "Area-based planning strategies" is an ambiguous scale. You need a clear definition.

10. Pg. 13, line 256-260: You can merge this paragraph into the following paragraphs about land use variables (Pg. 16, line 317-325).

11. I would also like to see more explicit acknowledgement that this is a cross-sectional that reveals associations but does not prove causation. The authors should be qualified in their language, and they should be careful to use wording that does not suggest a longitudinal study or causal relationships.

12. Other minor comments/typo/errors

Pg. 4, line 57: A typo: one 'However' should be deleted

Pg.12, line 237: the authors write: " Guangzhou is the largest city in Southern China and Provincial Capital". This is ambiguous statement. How to define the largest city? Area, population, or GDP? How about Shenzhen and Hongkong?

Pg. 13, line 250 and 252: “it defines a residential location” is ambiguous. This is a Chinese expression.

Pg. 13, line 259: “27349” -> “27, 349”

Pg. 13, line 266-267: in equation one, you need give a definition for ‘D’?

Pg. 15, line 299-300: ‘[14]’ -> ‘Rivera and Tiglao (2005)’

Pg. 18, line 346-347: “there is no need to measure the built environment from adjacent cells again” why? Do you think the surrounding environment of adjacent cells will not affect the commuting of travelers?

Pg. 19, line 366: ‘bandwidth’ needs further explanation

Pg. 19, line 374-375: I know AIC, but what is AICc? Based on your statement: “difference between the two models is more than 3”. Does this mean the difference of AIC value between the two models is greater than 3?

Pg. 20, line 381, 382: This looks like two tables, and the significance should be marked in the table with * or something else.

Pg. 22, line 427: “servicesencourages” -> “services encourages”

Pg. 31, line 590-591: “In contrast, mixed development would worsen the traffic in the morning peak.” I think your research results may be difficult to support such a conclusion.

7. PLOS authors have the option to publish the peer review history of their article (what does this mean?). If published, this will include your full peer review and any attached files.

Reviewer #3: No

Reviewer #4: No

---

## [Author Response · Author response to Decision Letter 1]

22 Aug 2021

GENERAL COMMENTS TO THE EDITOR AND REVIEWERS:

We are pleased to resubmit our revised manuscript of PONE-D-20-19283, which entitles ‘Spatially heterogeneous and double-edged effect of built environments on the commuting distance: from home-based and work-based perspectives’. We appreciate thoughtful comments and constructive suggestions of the reviewers. 

We explain how we revised the paper based on each comment as outlined below. 

RESPONSE TO EDITOR:

1. There are lots of language issues, still. For example, “rich literatures”, “commuting distances”, “built environment’s impact”, “considering…, conclusions”, Two “however in Line 57”, and so on. Please do find a native proofreading editor to go through the whole manuscript.

Response: Thanks for your careful work. Yes, we have invited an native editor to do the proofreading through the manuscript. 

2. Similarly, some statements are vague and confusing. Like the paper title, which features of commuting do you focus on? Volume of commuters? Commuting means a lot, it could be distance, time, volume or more. Be specific.

In the abstract, “the analysis generally assumes…”, such a statement is not academically rigorous, because it is not a “general” truth in built environment-travel studies. “To fill in the research gaps…” what gaps? It needs to be specific. Also, what are “home-based” and “work-based perspective”? They may be explained in the manuscript, but it confuses the readers without much context in the Abstract. I cannot point out all the academic writing issues here, but you should rewrite some sentences in the whole manuscript to make each sentence clear and connected with context.

Response: We changed the title as ‘Spatially heterogeneous and double-edged effect of built environments on the commuting distance: from home-based and work-based perspectives’.

Response: We changed the title as ‘Spatially heterogeneous and double-edged effect of built environments on the commuting distance: from home-based and work-based perspectives’. 

We revised the abstract part as: ‘Linear models assume that the influence of the built environment is spatially homogeneous. However, given the spatial heterogeneity of urban space, conclusions might be different or even be contrary. The influence of the built environment might also be different by home and work locations. To explore the spatially heterogeneous effect of the built environment from both home-based and work-based perspectives, this study applied large-scale cellular cellphone data in Guangzhou, China.’

3. As mentioned by one reviewer, the definition of home-based and work-based commuters are weird. Each commuter is both a resident and an employee. So how can we say “residents and employees” are “two distinct groups of commuter” (Line 80)? Are they just similar to the concepts of Trip attraction and trip departure in traditional transport literature.

Now I see your definition of “commuting” as “decay parameters of probabilistic distributions of commuting distances” what is P,D in Equation (1), how beta is estimated is unclear, some examples to interpret beta could be helpful. Do you need to standardize the data since you use an estimated parameter to represent commuting. So another issue here is how to define beta as what kind of commuting feature in your study. You can’t generally say it as “commuting”, as pointed out in Comment #2. Why not defining clearly what beta means, then using the a very clear concept to replace the vague statement of “commuting”. If I don’t misunderstand your definition, you actually look at the probability of long-distance commuting departing versus arriving at a zone (spatial cell), right? Is that what the “home-based” versus “work-based” commuting represents?

Response: Thanks for your comments. The definition of home-based and work-based commuting is determined how the commuting distance is aggregated into a spatial unit. In aggregate analysis, the commuting distance of a spatial unit is generally measured by the average value of all travelers’ commuting distances within that unit. It is important to note that, for the same spatial unit or neighborhood, there are two ways of averaging the commuting distance: home-based measure and work-based measure. The former measures the average travel distance of commuters who depart from the spatial unit, while the latter is based on commuters who arrive at the spatial unit. Because of the results of two measurements are different, it is necessary to differentiate between the home-based and work-based commuting distances. Current studies are limited in not considering the relations from home-based and work-based perspectives simultaneously. In addition, this study used the concept of ‘distance decay parameter’ instead of the average distance. 

We replaced D by X in the equation (1) to make it consistent with a general form of CDF: P(X<=x). Beta is estimated by a maximum likelihood method. The likelihood is given by eq.2. 

The concept of beta is explained as follows (lines 353-361): beta is estimated by maximizing L(beta,x) and we get: beta=mean(x). It implies that the estimated beta value equals the mean of observed samples of commuting distances. Despite that, measuring the commuting distance by a decay parameter cannot be simply seen as averaging all commuting distances in a spatial unit. The decay parameter beta represents the travel distance distribution of all commuters. For example, small beta value indicates a higher probability of shorter commuting distance and lower probability of longer commuting distance, whilst large beta value indicates a higher probability of longer commuting distance and lower probability of shorter commuting distance. The decay parameter also represents the expectation of the distance distribution in an exponential distribution. beta is not simply a mean of commuting distances. Rather, we should treat it as a weighted average of that, according to the property of an exponential distribution. In other words, the expectation value also shapes the slope of the curve. It represents either a high probability of longer distances and a low probability of shorter distances (large beta), or a high probability of shorter distances and a low probability of longer distances (small beta).

4. Related to the conceptualization of heterogeneity, or the “contextual effect”, you may refer to Zhang and Zhang 2018a,b in JPER and Urban Studies.

Response: Thank you for providing these critical references. We added them in the introduction part (lines 58-62): ‘Among these reasons, the spatial context is the most important one since it has both direct and indirect effects on travel. The direct effect means that built environment is treated as one of influencing factors along with transportation services and social demographic factors, such as in linear models. The indirect or moderating effects include multiplicity, interaction and scalability [7,8].’

7. Zhang M, Zhang W. When Context Meets Self-Selection: The Built Environment–Travel Connection Revisited. J Plan Educ Res. 2020;40: 304–319. 

8. Zhang W, Zhang M. Incorporating land use and pricing policies for reducing car dependence: Analytical framework and empirical evidence. Urban Stud. 2018;55: 3012–3033. 

RESPONSE TO REVIEWER #3:

1.According to reponse" home-based commuters who depart from the spatial unit,

while the latter is work-based commuters who arrive at the spatial unit", Usually a person starts from space unit A(home) to space B(work) , there is a problem: The same travel behavior is not only the starting point of space a, but also the arrival point of space B, should it be counted twice as both home and work based commuting?Another problem is that he work and resident at same space unit.How to Identify and differentiated two groups of commuters – residents and employees through the data processing, please give processing flow and why did you do that.

Response: Thanks for your careful review work. In aggregate analysis, the commuting distance of a spatial unit is generally measured by the average value of all travelers’ commuting distances within that unit. It is important to note that, for the same spatial unit or neighborhood, there are two ways of averaging the commuting distance: home-based measure and work-based measure. The former measures the average travel distance of commuters who depart from the spatial unit, while the latter is based on commuters who arrive at the spatial unit. Because of the results of two measurements are different, it is necessary to differentiate between the home-based and work-based commuting distances. Please check the statement in lines 87-95.

2.The independent variables need clarifications. How are dependent variables indicators?

Response: Yes, we added a description of independent and dependent variables in details in the section ‘Variables’. The dependent variables are the decay parameter beta in equation P(X<=x)=1-exp(-x/beta) (x>=0). It implies that the estimated beta value equals the expectation of the distance distribution in an exponential distribution. beta cannot be simply seen as averaging all commuting distances in a spatial unit. Rather, we should treat it as a weighted average of that, according to the property of an exponential distribution. In other words, the expectation value also shapes the slope of the curve. It represents either a high probability of longer distances and a low probability of shorter distances (large beta), or a high probability of shorter distances and a low probability of longer distances (small beta).

3.As the REVIEWER #2 suggest"Also, need to add a conceptual framework explaining why the relationships could be spatially varied". The figure added is so simple and cannont explain, more detailed influence action path and mechanism are needed.

Response: Thanks for your suggestion. We remade the conceptual framework (figure 1).

RESPONSE TO REVIEWER #4:

This is an interesting study, and I think the authors' data material and their findings of differentiated relation between land use and commuting in 4 different clusters can potentially result in a published article of high importance. However, in its present form, the paper has some serious shortcomings that must be amended before I can recommend publication.

1. For one thing, the author uses two ambiguous concepts (i.e. home-based commuting, work-based commuting). Of course, I think I understand what the meaning is here; each spatial unit may become a residential area or workplace area, depending on the direction of commuting. But why not directly use the residential area and workplace area here, or you can define a spatial unit to have both residential area and workplace area attributes. In short, ‘home-based commuting’ and ‘work-based commuting’ are not general academic terms.

Response: Thanks for your careful review work. In aggregate analysis, the commuting distance of a spatial unit is generally measured by the average value of all travelers’ commuting distances within that unit. It is important to note that, for the same spatial unit or neighborhood, there are two ways of averaging the commuting distance: home-based measure and work-based measure. The former measures the average travel distance of commuters who depart from the spatial unit, while the latter is based on commuters who arrive at the spatial unit. Because of the results of two measurements are different, it is necessary to differentiate between the home-based and work-based commuting distances. Please check the statement in lines 87-95.

2. In Literature review section, the author needs to add some literature about the relation between built environment of residential area and workplace area, and the commuting. You can easily search for them on the google scholar. This is the most relevant research to your research. Further, you may find that your research question is not a huge gap.

Response: Yes, we added a new literature review part about the relation between built environment and commuting distances (lines 139-201). 

3. In the framework, the author does not control any socioeconomic attributes, which may cause inaccurate results. For aggregated data, both statistical yearbooks and censuses can provide basic socioeconomic attributes in Chinese city, such as GDP, gender ratio, and elderly population ratio. Second, this research used a “5Ds” framework, but ignored Design and Distance to transit. Why? Because public transit accessibility, such as distance to bus/subway station or density of bus/subway station, is generally considered to be a very important factor affecting commuting. I think the author needs a very convincing reason here or needs to re-examine the framework.

Response: We did not include the socioeconomic attributes in the model. Our reason is that ‘This study does not include socioeconomic attributes, such as GDP, gender ratio, or elderly population ratio. The aim of this analysis is to explore the relation between the built environment and the commuting distance. By revealing the relation, the government can directly implement planning and policy measures for the built environment to address the problem of long commuting distances.’ (lines 382-386)

Thanks for your suggestion about Design and Distance to transit. We added two new variables: road intersections and bus stops. Based on that, we did new analysis using GWR and two-steps cluster models. 

4. For Variable Section, functional facility density belongs to Diversity rather than the Density (Pg. 15, line 295, 304-306; Ewing and Cervero 2010). Second, why choose Zhujiang New Town as the city center, and is Guangzhou a monocentric city? (Pg. 17, line 332-334). However, based on your statement: “The provision of housing encourages employees in the university to live near their workplaces, so the jobs-housing would be balanced. The work population decreases commuting distance in suburban areas (Pg. 22, line 413-415)”. In traditional monocentric city research, home-work distance is positively correlated with the distance to the city center. If you find that there is a low commuting distance area in the suburbs of Guangzhou, then I think Guangzhou may not be a monocentric city. So, you may need to re-examine how many city centers there are in Guangzhou. Besides, Closeness is not a general built environment variable; why use Closeness rather than road network distance to city center.

Response: According to the definition of density, it refers to ‘the variable of interest per unit of area (Ewing and Cervero, 2010)’. We think a single functional facility’s number still belongs to density, because it is measured by each type separately. We also have a diversity measurement, which calculates the POI mixture of all types.

Based on our previous analysis, we found that Guangzhou is a mono-centric city, and the center is Zhujiang New Town (Zheng et al. 2021). Although the work population decreases commuting distance in suburban areas, the city is still a monocentric city.

Closeness is actually a measurement of the centrality of the road network. A road with the highest closeness means that it is close to all roads in the study area, and it is the geometry center of road networks. We do not use the road network distance to center, because it is too similar with the distance to city center variable.

5. Method section lacks an introduction to Two-step cluster models.

Response: Thanks for your suggestion. We added an introduction of the two-step cluster model (lines 467-474): ‘To reveal the double-edged effect explicitly, a two-step cluster model is applied to further examine the area-based impact of the built environment and answer Research Question 3. The algorithm of the two-step cluster can be referred to Rundle-Thiele et al. [48]. In brief, the first step pre-clusters the data into several small sub-clusters by a cluster feature tree, and the second step aggregates sub-clusters into clusters by the standard hierarchical clustering algorithm. The clustering algorithm could generate different number of clusters. The optimal number of clusters is determined by Schwarz’s Bayesian information criterion (BIC).’

6. I suggest that the author add an introduction map of the research area, may be in the Study area section, otherwise it is difficult for readers to understand where the airport suburb, university town, and Huangpu suburb are.

Response: Thanks for your suggestion. We added a map of the study area (Figure 2). 

7. Pg. 3, line 55: the authors write: "Among these reasons, the spatial context is the most important one". Why? You need some evidence.

Response: The spatial context is import because it is a scientific issue which is not fully discussed in literature. Issues of built environment measurement and data source are relatively less important than the spatial context since they are method and technical issues which can be solved by improving the model. Studies from Zhang and Zhang (2018, 2020) confirm the opinion. We state it in the texts that ‘Among these reasons, the spatial context is the most important one since it has both direct and indirect effects on travel. The direct effect means that built environment is treated as one of influencing factors along with transportation services and social demographic factors, such as in linear models. The indirect or moderating effects include multiplicity, interaction and scalability [7,8]. These components cannot be directly measured by linear components.’ (lines 58-63)

8. Pg. 5, line 86: the authors write: "Home-based commuting represents the morning traffic peak and work-based commuting represents the evening traffic peak". I don’t agree, because commuting is two-way. Both of residential area and workplace area are very congested in the morning and evening.

Response: Yes, we realize your comment is correct. We remove the statement.

9. Pg. 5, line 98: "Area-based planning strategies" is an ambiguous scale. You need a clear definition.

Response: We revised it into ‘zonal planning’. 

10. Pg. 13, line 256-260: You can merge this paragraph into the following paragraphs about land use variables (Pg. 16, line 317-325).

Response: Yes, we merged them. Thanks for your suggestion.

11. I would also like to see more explicit acknowledgement that this is a cross-sectional that reveals associations but does not prove causation. The authors should be qualified in their language, and they should be careful to use wording that does not suggest a longitudinal study or causal relationships.

Response: Yes, this is a cross sectional study that reveals the associations.

12. Other minor comments/typo/errors

Pg. 4, line 57: A typo: one 'However' should be deleted

Response: Yes, we deleted one. 

Pg.12, line 237: the authors write: " Guangzhou is the largest city in Southern China and Provincial Capital". This is ambiguous statement. How to define the largest city? Area, population, or GDP? How about Shenzhen and Hongkong?

Response: We changed the statement as ‘the study area is the inner city of Guangzhou, China. Guangzhou is one of the four first-tier cities in China and Provincial Capital.’ 

Pg. 13, line 250 and 252: “it defines a residential location” is ambiguous. This is a Chinese expression.

Response: We changed it as ‘the location is defined as his/her residential place’.

Pg. 13, line 259: “27349” -> “27, 349”

Response: Yes. 

Pg. 13, line 266-267: in equation one, you need give a definition for ‘D’?

Response: We re-edit the equation using x and X instead of d and D. P(X<=x) is the probability (CDF) that the travel distance X is less than value x, since the probability of a continuous random variable can be only expressed by a cumulative distribution function.

Pg. 15, line 299-300: ‘[14]’ -> ‘Rivera and Tiglao (2005)’

Response: Yes. 

Pg. 18, line 346-347: “there is no need to measure the built environment from adjacent cells again” why? Do you think the surrounding environment of adjacent cells will not affect the commuting of travelers?

Response: Travelers’ commuting is already affected by adjacent cells in a GWR model. Thus, for the variables themselves, there is no need to consider adjacent cells again. Otherwise it conflicts with the GWR model.

Pg. 19, line 366: ‘bandwidth’ needs further explanation

Response: Bandwidth is the search range of the model. A cell is affected by all other cells within the range of the bandwidth. Bandwidth can be manually chosen or determined by criteria such as cross-validation or Akaike information criterion (AIC). Please also check it in lines 461-465.

Pg. 19, line 374-375: I know AIC, but what is AICc? Based on your statement: “difference between the two models is more than 3”. Does this mean the difference of AIC value between the two models is greater than 3?

Response: AICc is AIC with a small sample correction. AICc = AIC + 2K(K + 1) / (n - K - 1) where K is the number of parameters and n is the number of observations. When the sample size is small, there is a substantial probability that AIC will select models that have too many parameters, i.e. that AIC will overfit. To address such potential overfitting, AICc was developed: AICc is AIC with a correction for small sample sizes. Please check it in lines 489-496.

Pg. 20, line 381, 382: This looks like two tables, and the significance should be marked in the table with * or something else.

Response: Yes, we have split it into two tables. Significant variables are shown in maps with color, insignificant results are in grey color.

Pg. 22, line 427: “servicesencourages” -> “services encourages”

Response: Yes.

Pg. 31, line 590-591: “In contrast, mixed development would worsen the traffic in the morning peak.” I think your research results may be difficult to support such a conclusion.

Response: We revised as ‘In contrast, mixed development is associated with long home-based commuting trips’.

---

## [Decision Letter · Decision Letter 2]

5 Oct 2021

PONE-D-20-19283R2The spatially heterogeneous and double-edged effect of the built environment on commuting distance: home-based and work-based perspectivesPLOS ONE

Dear Dr. Zhou,

Thank you for submitting your manuscript to PLOS ONE. After careful consideration, we feel that it has merit but does not fully meet PLOS ONE’s publication criteria as it currently stands. Therefore, we invite you to submit a revised version of the manuscript that addresses the points raised during the review process.

We look forward to receiving your revised manuscript.

Kind regards,

Wenjia Zhang

Academic Editor

PLOS ONE

Journal Requirements:

Additional Editor Comments (if provided):

The revision is significantly improved. Besides the comments from the two reviewers, the authors can further proofread and refine the language, and make it more succinct for readers (e.g., by separating long paragraphs into two and deleting some unnecessary sentences or conjunctions).

Reviewers' comments:

Reviewer's Responses to Questions

**Comments to the Author**

1. If the authors have adequately addressed your comments raised in a previous round of review and you feel that this manuscript is now acceptable for publication, you may indicate that here to bypass the “Comments to the Author” section, enter your conflict of interest statement in the “Confidential to Editor” section, and submit your "Accept" recommendation.

Reviewer #3: (No Response)

Reviewer #4: All comments have been addressed

2. Is the manuscript technically sound, and do the data support the conclusions?

Reviewer #3: Partly

Reviewer #4: Yes

3. Has the statistical analysis been performed appropriately and rigorously? 

Reviewer #3: I Don't Know

Reviewer #4: Yes

4. Have the authors made all data underlying the findings in their manuscript fully available?

Reviewer #3: No

Reviewer #4: Yes

5. Is the manuscript presented in an intelligible fashion and written in standard English?

Reviewer #3: No

Reviewer #4: Yes

6. Review Comments to the Author

Reviewer #3: 1.The logic of introduction is not very smooth，especially line106-112 jump out with other parts, I recommend to put in line 584 “Guidance for area-based planning policies”.

2.The line 818 “Conclusions and discussions” this part is too thin. Authors should focus on the comparison of the research results with other studies and the reasons for this result

3.As the REVIEWER #2 suggest "Also, need to add a conceptual framework explaining why the relationships could be spatially varied". The figure added is so simple and cannont explain, more detailed influence action path and mechanism are needed. Sad to say that nothing seems to have changed in the revised version.

4.Why such hypothesis is puts forward? From the current version, it just lists the achievements of others, rather than discuss the how built environment impact commuting distance.

5.Some other minor defects shows that the author is not serious enough, such as:

1)Response: We changed the title as ‘Spatially heterogeneous and double-edged effect of built environments on the commuting distance: from home-based and work-based perspectives’.

2)In part of RESPONSE TO EDITOR, response 2 repeated the following paragraph:

“Response: We changed the title as ‘Spatially heterogeneous and double-edged effect of built environments on the commuting distance: from home-based and work-based perspectives’.”

3)As REVIEWER #4:9response, Area-based planning strategies ere revised into ‘zonal planning”, but I have seen” Guidance for area-based planning policies” as obvious secondary title in line 584.

4)Line43 Long commuting “distances” should be “distance”.

6. The authors use exactly the same paragraph replied to REVIEWER #3: 1 and REVIEWER #4: 1. Although the two comments are similar to some extent, the authors should still give a targeted answer.

Reviewer #4: The authors have made considerable improvements followed up my comments. However, several indicator classification error in the new text should be corrected.

1. In the most land use-travel researches, socioeconomic attributes are often used as the control variables in the model. One important reason is that you need to exclude the potential influence of the difference of socioeconomic attributes on the land use-travel relationships. In other words, if you do not control the socioeconomic attributes, the result can be biased. Therefore, you should at least clarify this limitation in the Discussion.

2. In 5Ds framework, Density measures the intensity of human activity per areal unit, such as population density and job density, rather than all the variables named density (Ewing and Cervero 2010). Diversity measures pertain to the number of different land use in a given area and the degree to which they are represented in land area, floor area, or employment (Ewing and Cervero 2010). So, functional facility density is closer to Diversity rather than Density.

3. Besides, space syntax closeness does not seem to belong to Destination accessibility. Destination accessibility measures ease of access to trip attractions, such as distance to employment center, distance to shopping center, and distance to city centers. However, space syntax closeness measures the form of network. So, space syntax closeness looks closer to Design, you may need some literature to support this variable.

4. Pg. 21, line 420: Public transit accessibility → Distance to transit

5. Please list the corresponding manuscript changes in quotation marks under the reviewer’s questions, if necessary.

7. PLOS authors have the option to publish the peer review history of their article (what does this mean?). If published, this will include your full peer review and any attached files.

Reviewer #3: No

Reviewer #4: No

---

## [Author Response · Author response to Decision Letter 2]

11 Dec 2021

Additional Editor Comments (if provided): The revision is significantly improved. Besides the comments from the two reviewers, the authors can further proofread and refine the language, and make it more succinct for readers (e.g., by separating long paragraphs into two and deleting some unnecessary sentences or conjunctions)

Response: Thanks for your suggestion. We made another round of proofreading. We improved the language in several sentences (line 53,81,87,112,159,173),And there are other revisions on improving the sentences. Please see them in the manuscript with track, at lines 239, 274, 275, 297, 230, 304, 309-310, 313-314, 321-325, 329-332, 348-349, 363, 413, 610-611, 618-619, 629, etc.

RESPONSE TO REVIEWER #3: 

1.The logic of introduction is not very smooth，especially line106-112 jump out with other parts, I recommend to put in line 584 “Guidance for area-based planning policies”.

Response: Thanks for your suggestion. We moved the paragraph into line 592. 

2.The line 818 “Conclusions and discussions” this part is too thin. Authors should focus on the comparison of the research results with other studies and the reasons for this result.

Response: Thanks for your suggestion. Yes, we added a new part to the conclusions and discussions. We compared our results with other studies. We emphasized the difference part and discussed the reason. The findings of the relationship between the built environment and the commuting distance are different from study to study. The reason is that these studies are based on different theoretical framework. For example, based on the co-location theory and individual choice mechanism, the co-location of jobs and housing reduces the commuting distance. However, other researchers have different opinions. In our study, we gave an explanation. In residential areas where the individual choice is the dominate mechanism, the co-location theory is convincing, but in business centers where the market mechanism is dominating, the co-location of jobs and housing has little effect on the home-based commuting distance. Industrial firms have much stronger bidding ability than individuals. Thus, individuals do not have free choices of housing locations at the business center. Similarly, it also explains why mix development increases the home-based commuting distance. Please see the lines 701–746 for details. 

3.As the REVIEWER #2 suggest "Also, need to add a conceptual framework explaining why the relationships could be spatially varied". The figure added is so simple and cannont explain, more detailed influence action path and mechanism are needed. Sad to say that nothing seems to have changed in the revised version.

Response: We appreciate this suggestion very much. After re-considering the theory behind our research findings, we made a new conceptual framework:‘It assumes that there are two mechanisms which dominate the relationship between the built environment and the commuting distance: the market mechanism and the individual choice mechanism. The relationships are varied at different urban locations, causing spatially heterogeneous effect. Also, the market mechanism and the individual choice mechanism are the leading force of the work-based and home-based commuting, respectively.’ (lines 126-130)

4.Why such hypothesis is puts forward? From the current version, it just lists the achievements of others, rather than discuss the how built environment impact commuting distance.

Response: Thanks for your comment. This question is related to the conceptual framework. We added new statement about the hypothesis underlying the spatially heterogeneous effect: ‘The debate on the impact of the built environment is mainly caused by the different mechanisms of human and urban space interaction: the market mechanism and the individual choice mechanism. The market mechanism triggers agglomeration economic and shapes the urban spatial structure. Economic agglomeration refers to a large number of firms existing in spatial proximity and benefit from cost reductions and efficiency gains. It encourages capital facilities and buildings to be concentrated located. The individual choice mechanism means a decision maker chooses the residential and work location with the highest utility from a set of alternatives. It assumes that workers choose home locations as close to their jobs as possible. The two mechanisms have different actions on different urban locations. The market mechanism is more competitive at business centers, and the choice mechanism has stronger influence at residential centers and suburban areas. Since the urban space is heterogeneous, the relationship between the commuting distance and the built environment should be spatially varied.’ (lines 56 – 68)

And we also discussed the hypothesis behind the home-based and work-based commuting distance: ‘Because the results of the two measurements are different, it is necessary to differentiate between home-based and work-based commuting distances. More importantly, the underlying mechanisms are different. From a work-based perspective, the economic agglomeration is the dominant mechanism. Industrial firms have much stronger land-rent bidding ability than individuals in a free market system. It forces local workers to live far from workplaces. From a home-based perspective, the co-location theory is the dominant mechanism. Workers can freely choose their home locations to avoid commuting time cost where the supply of housing land is adequate.’ (lines 99-107)

5.Some other minor defects shows that the author is not serious enough, such as:1)Response: We changed the title as ‘Spatially heterogeneous and double-edged effect of built environments on the commuting distance: from home-based and work-based perspectives’.2)In part of RESPONSE TO EDITOR, response 2 repeated the following paragraph:“Response: We changed the title as ‘Spatially heterogeneous and double-edged effect of built environments on the commuting distance: from home-based and work-based perspectives’.”3)As REVIEWER #4:9response, Area-based planning strategies ere revised into ‘zonal planning”, but I have seen” Guidance for area-based planning policies” as obvious secondary title in line 584.4)Line43 Long commuting “distances” should be “distance”.6. The authors use exactly the same paragraph replied to REVIEWER #3: 1 and REVIEWER #4: 1. Although the two comments are similar to some extent, the authors should still give a targeted answer.

Response: Thank you. This is a very useful suggestion that we learn from your comments. It’s true that the response should be targeted to different question. (for comments 1,2 and 6)

We revised ‘area-based planning’ to ‘zonal planning’ (for comments 3).

We revised ‘distances’ to ‘distance’ (for comments 4).

RESPONSE TO REVIEWER #4: 

The authors have made considerable improvements followed up my comments. However, several indicator classification error in the new text should be corrected.

1. In the most land use-travel researches, socioeconomic attributes are often used as the control variables in the model. One important reason is that you need to exclude the potential influence of the difference of socioeconomic attributes on the land use-travel relationships. In other words, if you do not control the socioeconomic attributes, the result can be biased. Therefore, you should at least clarify this limitation in the Discussion.

Response: Thanks for your comments. We discussed the limitation of not incorporating socioeconomic attributes in the new edition: ‘The study is limited in not incorporating individual socioeconomic attributes. Socioeconomic attributes are important factors which affect people’s commuting behavior. It is a common approach to explore the influence of people’s socioeconomic characteristic on the commuting distance, particularly in disaggregate analysis. In this study, we did not consider socioeconomic attributes as independent variables. Our argument is that government can implement spatial planning measures to decrease the commuting distance by improving the built environment, but socioeconomic attributes cannot be easily changed and are not effective policy measures for the government. Nevertheless, socioeconomic attributes still have potential influence on commuting behavior. Excluding them would cause biased results of the built environment and the commuting distance relationship. Realizing the shortcoming, we will incorporate individual-level data to further explore the behavioral drivers of commuting distance in future research.’ (lines 747-759)

2. In 5Ds framework, Density measures the intensity of human activity per areal unit, such as population density and job density, rather than all the variables named density

(Ewing and Cervero 2010). Diversity measures pertain to the number of different land use in a given area and the degree to which they are represented in land area, floor area, or employment (Ewing and Cervero 2010). So, functional facility density is closer to Diversity rather than Density.

Response: Thanks for your comments. We made revisions according to your comment. Please check them in lines 409-422.

3. Besides, space syntax closeness does not seem to belong to Destination accessibility. Destination accessibility measures ease of access to trip attractions, such as distance to employment center, distance to shopping center, and distance to city centers. However, space syntax closeness measures the form of network. So, space syntax closeness looks closer to Design, you may need some literature to support this variable.

Response: In our opinion, Closeness is still a destination accessibility index. Closeness in space syntax refers to the centrality level of road. Closeness (also normalized as syntactic ‘Integration’ (Hiller, 2010)) is a key index of the centrality. It indicates the accessibility and centrality level of spatial units (Li et al., 2019). In other words, it means the closeness of any given road section to all other road sections in the system (Hiller, 1984). Traditionally destination accessibility is measured by the distance to the city center. An underlying problem is that sometimes a city center is arbitrarily selected according to experience and subjective perception of a city. The index of ‘closeness’ solves the problem. It does not predefine a center. Rather, it measures a road section’s overall ease of moving to all other sections. Therefore, a location with higher closeness value has better destination accessibility. We also stated it in the manuscript at lines 426-436.

References:

Hillier B, Stonor T. Space syntax - strategic urban design. City Planning Institute of Japan. 2010. 

Li Q, Zhou S, Wen P. The relationship between centrality and land use patterns: Empirical evidence from five Chinese metropolises. Computers, Environment and Urban Systems. 2019;78. doi:10.1016/j.compenvurbsys.2019.101356

Hillier Bill, Hanson Julienne. The social logic of space. Cambridge University Press; 1984.

4. Pg. 21, line 420: Public transit accessibility → Distance to transit

Response: Yes, we revised it according to your suggestion: ‘Distance to transit can be alternatively measured by the number of stations per unit area, namely public transit accessibility. In this study, it is the number of bus stops in a grid.’ (lines 424-425)

5. Please list the corresponding manuscript changes in quotation marks under the reviewer’s questions, if necessary.

Response: This is really a useful suggestion. We list changes in quotation marks to make it easier for reviewing.

---

## [Decision Letter · Decision Letter 3]

5 Jan 2022

The spatially heterogeneous and double-edged effect of the built environment on commuting distance: home-based and work-based perspectives

PONE-D-20-19283R3

Dear Dr. Zhou,

We’re pleased to inform you that your manuscript has been judged scientifically suitable for publication and will be formally accepted for publication once it meets all outstanding technical requirements.

Kind regards,

Wenjia Zhang

Academic Editor

PLOS ONE

Additional Editor Comments (optional):

Reviewers' comments:

Reviewer's Responses to Questions

**Comments to the Author**

1. If the authors have adequately addressed your comments raised in a previous round of review and you feel that this manuscript is now acceptable for publication, you may indicate that here to bypass the “Comments to the Author” section, enter your conflict of interest statement in the “Confidential to Editor” section, and submit your "Accept" recommendation.

Reviewer #4: (No Response)

2. Is the manuscript technically sound, and do the data support the conclusions?

Reviewer #4: (No Response)

3. Has the statistical analysis been performed appropriately and rigorously? 

Reviewer #4: (No Response)

4. Have the authors made all data underlying the findings in their manuscript fully available?

Reviewer #4: (No Response)

5. Is the manuscript presented in an intelligible fashion and written in standard English?

Reviewer #4: (No Response)

6. Review Comments to the Author

Reviewer #4: The authors have, in my opinion, responded to and followed up my comments in a satisfactory way. This paper could be accepted for publication.

7. PLOS authors have the option to publish the peer review history of their article (what does this mean?). If published, this will include your full peer review and any attached files.

Reviewer #4: No

---

## [Editor Report · Acceptance letter]

22 Feb 2022

PONE-D-20-19283R3 

The spatially heterogeneous and double-edged effect of the built environment on commuting distance: home-based and work-based perspectives 

Dear Dr. Zhou:

I'm pleased to inform you that your manuscript has been deemed suitable for publication in PLOS ONE. Congratulations! Your manuscript is now with our production department. 

Kind regards, 

on behalf of

Dr. Wenjia Zhang 

Academic Editor

PLOS ONE